# Repo2Run: Automated Building Executable Environment for Code Repository at Scale

Ruida Hu[1][*]    Chao Peng[2][†]    Xinchen Wang[1][*]    Junjielong Xu[2]    Cuiyun Gao[1][†]

[1]Harbin Institute of Technology, Shenzhen      [2]ByteDance, Beijing
{200111107, 200111115}@stu.hit.edu.cn
{pengchao.x, xujunjielong.l}@bytedance.com   gaocuiyun@hit.edu.cn

## Abstract

Scaling up executable code data is significant for improving language models' software engineering capability. The intricate nature of the process makes it labor-intensive, time-consuming, and expert-knowledge-dependent to build a large number of executable code repositories, limiting the scalability of existing work based on running tests. The primary bottleneck lies in the automated building of test environments for different repositories, which is an essential yet underexplored task. To mitigate the gap, we introduce Repo2Run, the first LLM-based agent aiming at automating the building of executable test environments for any repositories at scale. Specifically, given a code repository, Repo2Run iteratively builds the Docker image, runs unit tests based on the feedback of the building, and synthesizes the Dockerfile until the entire pipeline is executed successfully. The resulting Dockerfile can then be used to create Docker container environments for running code and tests. We created a benchmark containing 420 Python repositories with unit tests for evaluation. The results illustrate that Repo2Run achieves an 86.0% success rate, outperforming SWE-agent by 77.0%. The resources of Repo2Run are available at `https://github.com/bytedance/Repo2Run`.

## 1   Introduction

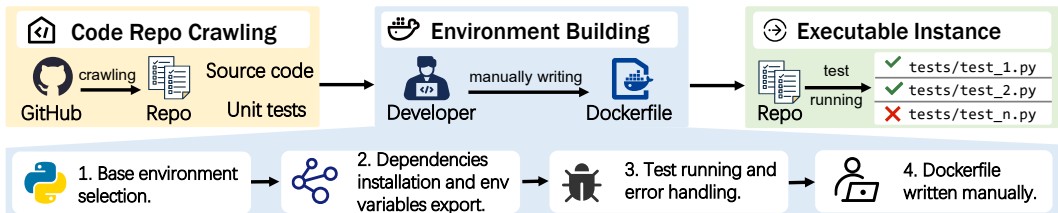

Figure 1: The pipeline of code repository mining and manual environment building. Developers manually write Dockerfiles through iterative steps including base environment selection, dependency installation, test running, error handling, validating the environment by running unit tests.

Large language models (LLMs) have recently illustrated significant progress on solving software engineering issues [1], driving the advent of numerous coding LLM agents like MetaGPT [2], SWE-agent [3], OpenHands [4], Copilot [5], and Cursor [6]. However, the lack of training environment

---

[*]Work done during an internship at ByteDance.
[†]Corresponding authors.

39th Conference on Neural Information Processing Systems (NeurIPS 2025).

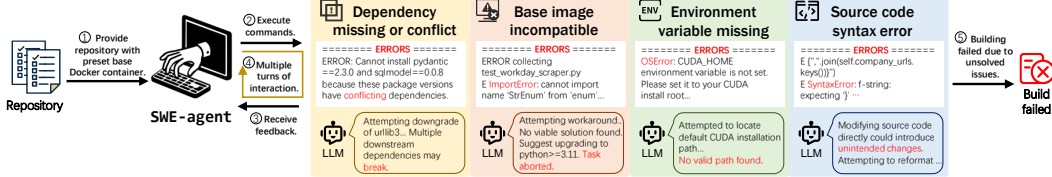

Figure 2: Four error types (highlighted in different colors) that SWE-agent fails to resolve during environment building.

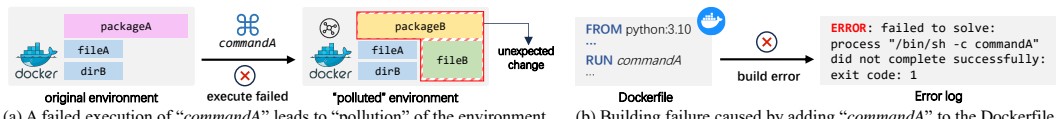

(a) A failed execution of "*commandA*" leads to "pollution" of the environment.    (b) Building failure caused by adding "*commandA*" to the Dockerfile.

Figure 3: The illustration of a command executing failed and "polluting" the environment. (a) Failed commands like "*commandA*" can irreversibly "pollute" the environment by altering packages, files, or directories, making subsequent builds unstable. (b) To reproduce the changes, such a failed command "RUN commandA" needs to be added to the Dockerfile. However, adding it will lead to building failure.

remains a significant challenge to advancing LLMs in software engineering — As shown in Figure 1, while static code is readily accessible through the GitHub API, building *executable environments* for code testing and patch validation demands extensive manual effort even for skilled developers [3]. This forces researchers to choose between (1) relying on unreliable static code metrics for rewarding (e.g., SWE-Fixer [7]) and SWE-RL [8], and (2) engaging developers to manually build limited repository environments (e.g., SWE-Gym [9] and SWE-Smith [10]) without scalability. Thus, an automated environment building infrastructure is urgently needed but remains underexplored.

In this paper, we introduce Repo2Run, an LLM-based agent designed to build executable environments. The key idea is that if the agent can successfully navigate the process of building a repository's dependencies and running its unit tests, its actions can be recorded and replayed to programmatically synthesize a runnable Dockerfile. This enables automated creation of isolated and consistent environments across different platforms via only one line command "docker build", eliminating "*it works on my machine*" problem [11].

Repo2Run addresses two key challenges in automating executable environment building:

1. **Hard to explore a valid trajectory that can successfully build the environment.** We found that without effective actions, the LLM agent would struggle to handle such a complex process. As shown in Figure 2, SWE-agent [3], a well-known LLM-based agent, often struggles with unresolved issues during environment building due to its lack of specialized actions. To address this challenge, Repo2Run uses an internal environment for building and an external environment for assistance, with nine actions designed to resolve these issues.

2. **Failing to synthesize a runnable Dockerfile without execution failure.** As shown in Figure 3, we observed that some incorrect commands in the trajectory can "pollute" the environment, leading to errors during synthesis. To tackle this challenge, we design an adaptive rollback action. When encountering failed commands like "*commandA*" that may cause "pollution" to the environment, Repo2Run restores the internal environment to its pre-execution state. This ensures the building process remains consistent and repeatable using the Dockerfile, avoiding unexpected errors.

To demonstrate Repo2Run's effectiveness, we first created a benchmark of 420 latest Python repositories with unit tests from GitHub in 2024, as existing datasets include only a very small number of repositories [1]. We choose Python for two key reasons: (1) the most popular software engineering benchmarks (e.g., SWE-bench [1]) focus on Python repositories, and (2) Python is one of the most widely used programming languages today [12]. We then evaluated Repo2Run on this benchmark, finding it successfully built the environments for 361 repositories, achieving an 86.0% success rate,

---

[3]We conducted a study to show how industry developers struggle in building environments. See Appendix F.

which is 63.9% higher than the leading baseline. Notably, Repo2Run outperforms the SWE-agent by 77.0%, demonstrating the effectiveness of the proposed agent.

We believe Repo2Run will serve as the foundational infrastructure, driving future research at the intersection of AI and Software Engineering and facilitating the community to efficiently scale up their executable code data and environments.

## 2 Formulation

In this part, we define the task of executable environment building. Given a repository $R$, executable environment building aims to determine a suitable **base image** $B$ and a **building process** $\mathcal{P}$ such that the resulting environment state $S$ satisfies the verification $\varepsilon$, as defined in Equation (1):

$$\varepsilon(R, S) = 0, \text{where } S = \delta(B, \mathcal{P}) \tag{1}$$

Here, $\delta$ is the state transition function, and $\varepsilon(R, S) = 0$ indicates that the verification $\varepsilon$ is satisfied.

### 2.1 State transition

• **Environment state** ($S$): The environment state $S \in \mathcal{S}$ represents the current state of the computer system, which encompasses all variables, files, cache, etc.

• **Command sequence** ($\mathcal{C}$): The command sequence $\mathcal{C} \in \mathbb{C}$ represents a set of individual commands. Each individual command $C \in \mathcal{C}$ refers to an instruction or action that can be executed in the environment via interfaces like `bash`, thereby changing the system state.

• **State transition** ($\delta$): The state transition $\delta$ is a function that defines the process through which the system transitions from the start state $S_{start} \in \mathcal{S}$ to the end state $S_{end} \in \mathcal{S}$ after executing of a command sequence $\mathcal{C}$, as defined in Equation (2):

$$\delta : \mathcal{S} \times \mathbb{C} \to \mathcal{S}, \quad \delta(S_{start}, \mathcal{C}) = S_{end} \tag{2}$$

### 2.2 Base image

• **Empty state** ($S_{\emptyset}$): The empty state $S_{\emptyset} \in \mathcal{S}$ represents a completely bare operating system or a purely hypothetical state without any builds.

• **Base image** ($B$): The base image $B \in \mathcal{S}$ is a special type of environment state, typically managed by professional teams for user convenience, such as "`python:3.10`". Starting from the empty state $S_{\emptyset}$, the base image $B$ can be created by executing a predefined command sequence $\mathcal{C}_B \in \mathbb{C}$, i.e., $B = \delta(S_{\emptyset}, \mathcal{C}_B)$. Users can utilize these base images by simply adding their names to the Dockerfiles.

### 2.3 Building process and state verification

• **Building process** ($\mathcal{P}$): The building process $\mathcal{P} \in \mathbb{C}$ is the command sequence designed to build the environment from the base image $B$. We denote the resulting state as $S_f \in \mathcal{S}$, where $S_f = \delta(B, \mathcal{P})$.

• **State verification** ($\varepsilon$): The state verification $\varepsilon$ is a Boolean function used to determine whether the resulting state $S_f$ successfully runs all tests in the repository $R$. $\varepsilon(R, S_f) = 0$ indicates that all tests in the repository $R$ can be successfully run. Otherwise, at least one test fails.

## 3 Repo2Run

In this part, we introduce the design of Repo2Run. Given a code repository, Repo2Run synthesizes a runnable Dockerfile, enabling the building of an executable environment. As shown in Figure 4, this process is divided into two phases: the build phase and the record phase.

In **build phase**, Repo2Run employs a dual-environment architecture consisting of an **internal environment** and an **external environment**. The internal environment is a Docker container where the agent can execute actions restricted to a container, e.g., test running. The external environment is a terminal where the agent can execute actions prohibited within a container, e.g., base image change.

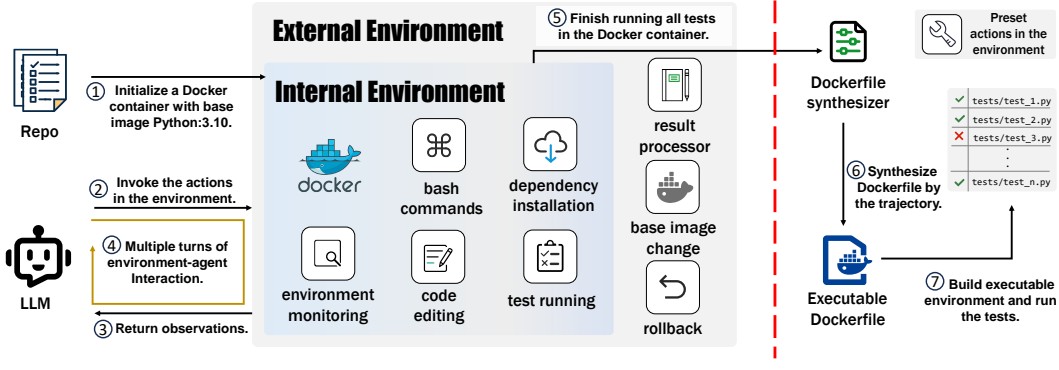

Figure 4: The workflow of Repo2Run, involving two phases: the build phase and the record phase. The build phase utilizes a dual-environment architecture: the **internal environment** with five actions for environment building, while the **external environment** with three actions assists the internal environment. The record phase converts the validated command sequence into a runnable Dockerfile for reconstructing the executable environment. See Appendix A for more examples of these actions.

Once all tests in the internal environment are successfully executed, Repo2Run transitions to the record phase.

In **record phase**, Repo2Run synthesizes a runnable Dockerfile based on the command sequence executed during the build phase. This Dockerfile serves as a precise record of the build process, and its execution can be viewed as a replay of the original build phase.

Next, we introduce the external environment, internal environment, and Dockerfile synthesizer.

## 3.1    External environment

The external environment serves as a bridge between the LLM and internal environment. It transmits actions from the LLM to the internal environment and relaying observations back. Following with the ReAct framework [13], the external environment maintains a history of thoughts, actions (i.e., successfully executed commands), and observations. This iterative process ensures that the LLM stays informed about the system's state and makes accurate decisions while effectively interacting with the internal environment.

• **Rollback**: As shown in Figure 3, commands like "*commandA*" [4] may "pollute" the environment, making it difficult to reproduce using a Dockerfile. When a command fails (i.e., returns a non-zero return code), the environment transitions to an uncertain state. Therefore, we introduce a rollback action to ensure the building process remains consistent. Specifically, before executing a command, we use "`docker commit`" to create a reserve snapshot of the current state (i.e., save the image at that point). If the return code is not 0, indicating a failure, the environment is rolled back to the most recent reserve snapshot. However, certain commands, referred to as safe commands (e.g., "`cat`"), generally do not alter the state of the environment and are exempt from rollback. For a complete list of safe commands, see Appendix B.2.

• **Base image change**: If the LLM agent determines that the current base image is unsuitable during the environment building process, it can reselect and switch to a new base image. This change invalidates the previous building process, clearing all executed commands and requiring a restart.

• **Result processor**: Command execution during interactions can produce extensive output (e.g., error logs), potentially overwhelming the LLM agent. To mitigate this, long outputs are truncated by this action, retaining only the initial and final sections up to a specified length.

---

[4]We collected some commands that can "pollute" the environment, see Appendix C.

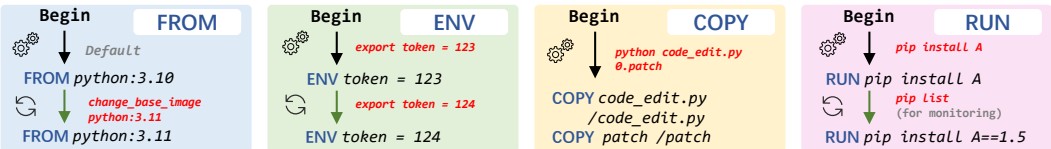

Figure 5: Rules for Dockerfile synthesis, illustrating how the Dockerfile synthesizer maps executed commands into Dockerfile statements using four keywords: "FROM", "ENV", "COPY", and "RUN". Black arrows represent the creation of statements, while green arrows indicate their transformations. Red text next to the arrows specifies the commands executed during each step.

## 3.2 Internal environment

The internal environment is a Docker container. Based on the latest data [14] from 2025, we select Python 3.10 as the default Docker base image due to its broadest adoption among Python versions.

• **Environment monitoring**: It serves as the eyes of the LLM agent in the internal environment, allowing it to observe the current environment state. These commands typically do not change the state of the environment. Basic commands like "ls" and "cat" are used to inspect directories and files, "find" is used to locate files, while more advanced commands like "pip list" and "pipdeptree" [15] retrieve the versions of installed libraries and dependency relationships.

• **Dependency installation**: It serves as installing third-party packages required for running tests, using tools like pip for Python and apt-get for system-level packages. To resolve potential conflicts (e.g., version constraints), a dependency management is implemented to assist the installation, where packages to be installed are added to a waiting list and resolved as needed. If installation succeeds, the environment is updated; otherwise, the rollback is performed. For detailed designs, see Appendix G.

• **Test running**: It serves as both a compass and a checkpoint, guiding the building process and verifying whether the environment runs all tests successfully. By executing "pytest" for unit tests, it determines whether the Docker container has been correctly built. If all tests run successfully, the process concludes; otherwise, error logs are sent to the LLM agent for further adjustments.

• **Code editing**: It enables the LLM agent to modify the code within the internal environment, including both inside and outside the repository. Direct code editing is rare, but sometimes needed for issues like syntax errors. To prevent the LLM agent from bypassing tests by directly altering or deleting test files, it is restricted from modifying or deleting the original test files within the repository.

• **Bash commands**: Like many other LLM-based agents [2–4], Repo2Run is allowed to invoke bash commands, allowing it fully operate within the Docker container. It enables the agent to execute dynamically defined actions that are not pre-specified, ensuring flexibility in handling various errors.

## 3.3 Dockerfile synthesizer

Once the LLM agent successfully runs all tests, the Dockerfile synthesizer converts the command sequence from the building process into a runnable Dockerfile. It processes each command sequentially, following the rules in Figure 5 to synthesize Dockerfile statements. The synthesis uses four Dockerfile keywords: "FROM", "ENV", "COPY", and "RUN". For more details and examples, see Appendix B.

• FROM: The FROM statement defines the base image and is typically the first line in a Dockerfile. If the LLM agent changed the base image, the FROM statement must be updated, and all subsequents are cleared to ensure consistency.

• ENV: The ENV statement persistently sets environment variables in the Docker container. When a command with "export" is detected, it is converted into an ENV statement. If the vaule of an environment variable is overwritten, this statement is updated accordingly.

• COPY: The COPY statement copies local files or directories into the Docker container. As shown in Figure 5, it is often used for code editing to import editing scripts (i.e., code_edit.py) and patches.

• RUN: The RUN statement executes commands in the container, with each line creating a new bash session. It cannot be used for persistent environment variables. Specially, for every installed packages, versions are recorded to ensure reproducibility, and the RUN statement is updated accordingly.

# 4 Experiment

We evaluate the effectiveness of Repo2Run on 420 Python code repositories. As the popular option, we select `gpt-4o-2024-05-13` for all experiments, with the temperature uniformly set to 0.2.

## 4.1 Benchmark

To the best of our knowledge, there is no prior work similar to Repo2Run that automates executable environment building. Existing manually constructed datasets are limited to very few repositories [1]. To validate the capability of Repo2Run, we create a new benchmark consisting of filtered Python repositories from GitHub based on the following criteria:

• **Creation date**: To avoid the data leakage, we select repositories created in 2024, ensuring they are not part of mainstream LLM training data.

• **Star count**: To maintain quality, we only include repositories with more than 100 stars.

• **Test directory**: We focus on repositories likely to contain unit tests, identified using `pytest`, a leading Python testing framework compatible with tools like `unittest`. Pytest detects test files with a "`test_`" prefix or "`_test`" suffix, typically located in "`test`" or "`tests`" directories. We only retain repositories that have these directories.

Using these criteria, we crawled 449 repositories in December 2024 and filtered 420 containing at least one unit test to form our benchmark. For statistic of their scale, see Appendix D.

## 4.2 Evaluation metrics

• **Dockerfile Generation Success Rate (DGSR)**: It indicates the percentage of attempts where the method successfully generates a **runnable** Dockerfile. To be considered successful, the generated Dockerfile must be able to build without errors. If the Dockerfile for a code repository successfully builds, it is regarded as a successful generation. Generating runnable Dockerfile is fundamental for successfully building the executable environment.

• **Environment Building Success Rate (EBSR)**[5]: It represents the percentage of attempts where the method successfully builds executable environments. For a successful building, the generated Dockerfile must not only build successfully but also allow tests to run by "`pytest`" in the Docker container. We are only concerned with whether tests can be executed, regardless of whether they pass or fail, as outcomes of tests may inherently vary within the repository.

## 4.3 Baselines

• **pipreqs** [16]: It is an automated tool that generates a "`requirements.txt`" file by analyzing the import statements in the Python scripts and identifying the necessary dependencies without LLM. Using the `requirements.txt` file generated by pipreqs, we create a Dockerfile. The detail is provided in Appendix I.1.

• **LLM generator**: The "`README`" file in a code repository usually contains environment building instructions. Therefore, we directly drive the LLM to read the "`README`" file and generate an executable Dockerfile accordingly.

• **SWE-agent** [3]: SWE-agent establishes a custom agent-computer interface (ACI) that uses the LLM agent's interaction with the repository environment by allowing actions such as reading files, editing files, and executing bash commands. Initially intended as an LLM agent for bug fixing, we preserve its framework and default settings, adjust its prompts, as shown in Appendix I.2.

## 4.4 Experimental Results

The results of different baselines are presented in Table 1. We observe that Repo2Run consistently outperforms other baselines on both DGSR and EBSR. Repo2Run ultimately completed environment building for 361 code repositories, achieving an EBSR of 86.0%. It is 63.9% higher than the highest rate achieved by other methods, demonstrating great advantages. Due to the design of

---

[5]We have also conducted a direct evaluation based on test pass rates in Appendix E.

Table 1: The results of different baselines, including the percentage and number of successfully generated Dockerfiles and successfully built environments.

| Metric | DGSR | # Successfully Generated Dockerfiles | EBSR | # Successfully Built Environments |
|--------|------|------|------|------|
| **pipreqs** | 29.8% | 125 | 6.0% | 25 |
| **LLM generator** | 47.6% | 200 | 22.1% | 93 |
| **SWE-agent** | 26.9% | 113 | 9.0% | 38 |
| **Repo2Run** | **100%** | **420** | **86.0%** | **361** |

Table 2: Repository distribution across different domains and their EBSR, where SE means Software Engineering, NLP means Natural Language Processing, and DV means Data Visualization.

| Type | AI / ML | SE | NLP | DV | Security | Others | All |
|------|---------|-----|-----|-----|----------|--------|-----|
| **# Success** | 221 | 70 | 41 | 10 | 7 | 12 | 361 |
| **# Total** | 267 | 73 | 47 | 11 | 8 | 14 | 420 |
| **EBSR** | 82.8% | 95.9% | 87.2% | 90.9% | 87.5% | 85.7% | 86.0% |

rollback, Repo2Run successfully generates Dockerfiles that can be built successfully for all 420 code repositories, which other actions cannot guarantee. For details of each repostories, see Appendix H.

For pipreqs, the main failures come from two reasons. First, generating the requirements.txt fails when there are issues within the repository, such as encoding errors or syntax errors in the files. This happens in 30 repositories (7.1%). Second, even when requirements.txt is generated, it might not be downloaded properly due to package version conflicts. This occurs in 265 repositories (63.1%). Besides, both the LLM generator and SWE-agent fail to ensure that the generated Dockerfile can be successfully built due to the lack of an ensuring mechanism. Surprisingly, the ability of SWE-agent, a general agent framework, to generate Dockerfiles is even weaker than simply letting the LLM read the "README" file. This indicates that a general agent framework cannot guarantee the generation of runnable Dockerfiles. Ensuring mechanisms like rollback is necessary to effectively use the interactive information from the agent to generate runnable Dockerfiles.

As shown in Table 2, we manually categorized the 420 repositories into six domains. EBSR across these domains all exceed 80%, outperforming all baselines and demonstrating Repo2Run's consistent performance across different domains. Prior research [17, 18] shows that the real-world benchmarks differs from synthetic ones and better reflect a model's true capabilities. Our benchmark is built entirely from real-world GitHub data, filtered only by repository creation date, star count, and test, with no additional filtering. Thus, it accurately represents the current distribution of Python repositories. Moreover, a previous real-world benchmark [18] includes 59.6% of AI/ML repositories, closely matching the 63.6% proportion in our benchmark. This alignment further confirms that our benchmark reflects the real-world scenarios.

## 4.5  Ablabtion of Repo2Run

To investigate the impacts of the dual-environment architecture and Dockerfile synthesizer separately, as two parts of Repo2Run, we separately remove each component of them. For the experiment without the dual-environment architecture, we retain only the internal environment's bash commands as the most basic interface and remove all other actions. For the experiment without the Dockerfile synthesizer, we directly instruct the LLM to synthesize a runnable Dockerfile.

Experimental result of the ablation study is shown in Table 3. We observe that removing the dual-environment architecture and retaining only bash commands results in a 7.6% decrease in DGSR. The main reason for this drop is the removal of rollback and other designs, making the system more prone to entering uncertain states and subsequently failing to reproduce. In addition, EBSR shows a 44.3% decrease, primarily because the simplification of design makes it more difficult for the LLM agent to execute all tests in the internal environment. Besides, removing the Dockerfile synthesizer directly leads to an 80.5% drop in DGSR. This indicates that having the LLM directly generate Dockerfiles is unlikely to fully follow the event history, resulting in Dockerfiles that fail to build successfully.

Table 3: The results of ablation experiments, including the percentage and number of successfully generated Dockerfiles and successfully built environments.

| Metric | DGSR | # Successfully Generated Dockerfiles | EBSR | # Successfully Built Environments |
|---|---|---|---|---|
| **w/o dual-environment** | 92.4% ↓ 7.6 | 388 | 41.7% ↓ 44.3 | 175 |
| ↪ w/o rollback | 96.9% ↓ 3.1 | 407 | 83.6% ↓ 2.4 | 351 |
| **w/o Dockerfile generator** | 19.5% ↓ 80.5 | 82 | 13.8% ↓ 72.2 | 58 |
| **Repo2Run** | 100% | 420 | 86.0% | 361 |

This also directly causes a sharp decline in EBSR. Specially, we also perform an ablation study for the rollback. The results show that removing the rollback causes 3.1% of generated Dockerfiles to become unrunnable, demonstrating the effectiveness of the rollback mechanism.

It is also observed that Repo2Run without the Dockerfile synthesizer is outperformed by the LLM generator. This is because the LLM generator leverages the "README" file, which provides a clear, simple and high-level overview of the executable environment building, allowing for more accurate Dockerfile synthesizer. In contrast, the event history-based approach lacks this context, making it harder for the LLM to fully understand the building goals. However, the Dockerfile synthesizer effectively utilizes the detailed event history, highlighting the complementary roles of both components of Repo2Run in generating a reliable runnable Dockerfile.

# 5 Discussion

## 5.1 Time consumption

Figure 6 shows the distribution of time spent successfully building 361 code repositories. The average time for successful building using Repo2Run is 29.03 minutes. 111 (30.7%) of the repositories are successfully built in less than 10 minutes. Additionally, our empirical study through sampling indicates an average manual building time of 21.33 minutes (Appendix F). Considering network differences and randomness, Repo2Run achieves a time consumption comparable to manual building. Additionally,

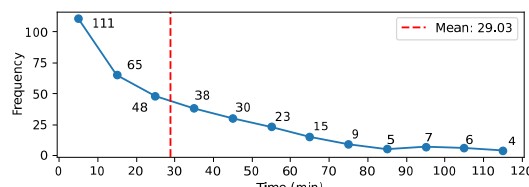

Figure 6: Time distribution of successful building.

for complex issues, Repo2Run shows greater advantages over manual building. Repo2Run successfully builds all the cases that were manually successful in our empirical study.

Table 4: Analysis of failed cases in Repo2Run.

| Category | # Case (%) |
|---|---|
| Hardware Insufficiency | 21 (35.6%) |
| Missing Token | 1 (1.7%) |
| Repository Defects | 5 (8.5%) |
| Dependency Installation Timeout | 8 (13.5%) |
| Runtest Timeout | 24 (40.7%) |

Table 5: Comparison of Dockerfile quality. "Warning" counts in each group show Repo2Run vs. baseline.

| Baseline | # "Warning" |
|---|---|
| vs. pipreqs | 5.17 / 6.0 |
| vs. LLM generator | 5.42 / 10.16 |
| vs. SWE-agent | 5.43 / 6.49 |

## 5.2 Failure case study

For the code repositories that fail to build, we manually inspect the reasons for failures and find that most are due to issues within the repositories themselves. Table 4 demonstrates the analysis of 59 failed cases in Repo2Run. The first three categories (45.8%) primarily fail due to external factors, while the latter two categories (54.2%) are unable to be configured successfully within the allotted time due to excessive dependencies and extensive tests inherent to the repositories. We also provide a detailed case in Appendix J.

## 5.3 Dockerfile Quality

We utilize `hadolint` [19] to analyze the quality of the Dockerfile generated by Repo2Run, comparing with Dockerfiles successfully generated by three baselines and counting the number of warnings. Fewer warnings indicate higher quality in Dockerfile generation. As shown in Table 5, Dockerfiles generated by Repo2Run's consistently have fewer warnings than those generated by baselines. Specifically, compared to `pipreqs`, LLM generator, and SWE-agent, Repo2Run reduces warnings from 6.0 to 5.17, 10.16 to 5.42, and 6.49 to 5.43, respectively. The results indicate that Repo2Run produces higher-quality Dockerfiles with fewer potential issues.

## 5.4 Limitation

Repo2Run can automate building executable environments for many repositories but may still encounter challenges in some edge cases. For complex environments that remain unresolved, manual verification and intervention are necessary. Nonetheless, as an important step toward full automation, we believe this work makes a significant contribution to the AI community's infrastructure.

# 6 Related Work

## 6.1 LLM-based agent

LLM-based agents [20, 21] typically consist of four key components: planning, memory, perception, and action [22]. Planning is crucial for agent systems, as it schedules agents to ensure a smooth process. LLM-based agents employ various planning strategies, including single [23] or multiple planners [24], single [25] or multi-turn planning [26], and single [27] or multi-path planning [28]. The memory component in LLM-based agents stores historical data to support coherent reasoning and complex tasks. Implementations vary in terms of memory duration (short-term [29] or long-term memory [30]), memory ownership (specific [31] or shared memory [32]). For perception, LLM-based agents primarily utilize textual input [33, 30] (natural language and programming language) and visual input [34, 35] (images and diagrams) to perceive and process information. To extend capabilities beyond interactive dialogue, the action component employs various external tools [36, 37], such as searching tools, file operations, and GUI operations. Nowadays, LLM-based agents have demonstrated superior performance compared to standalone LLMs in various software engineering tasks [38–47]. Many complex software engineering problems, especially repository-level challenges [48–51], can thereby be effectively addressed. However, no LLM-based agents are specifically designed for executable environment building currently. To fill this gap, this paper employs a novel approach for automated executable coding environment building and runnable Dockerfile generation.

## 6.2 Environment building and Dockerfile generation

There have already been many efforts devoted to environment building. Oss-Fuzz-Gen [52] relies on predefined build instructions (such as "`./bootstrap.sh`", "`./configure`", "`make`") to build projects for fuzzing but lacks of flexibility for diverse projects when specified files are absent. Existing solutions that help developers write Dockerfiles broadly fall into three categories: (1) Template-based generators that create Dockerfiles based on project context [53, 54], (2) Task-specific tools such as pipreqs [16], DockerizeMe [55] which supports environment dependency inference for Python projects, and DockerGen [56] for dependency recommendations based on knowledge graphs built from existing Dockerfiles, whose overall effectiveness still has substantial room for improvement [57], and (3) Code completion tools including GitHub Copilot [5] and HumpBack [58] that generate suggestions for developers while writing Dockerfiles. The use of deep learning models to generate Dockerfiles based on natural language specifications of software requirements is also investigated [59]. While these approaches provide valuable assistance, they either require significant manual input from developers or are limited to specific use cases.

# 7 Conclusion

In this paper, we propose Repo2Run, the first LLM-based agent for automated coding executable environment building and Dockerfile generation for Python repositories. With a dual-environment architecture and a Dockerfile synthesizer, Repo2Run is able to select and change the base image, manage and install dependencies based on action observation and rollback mechanism, and utilize bash commands and existing test suites. Our evaluation of 420 popular Python repositories hosted on GitHub demonstrates the effectiveness of Repo2Run with an 86.0% success rate. We believe Repo2Run will serve as the foundational infrastructure, enabling the community to efficiently scale up executable environments.

# 8 Acknowledgement

This research is supported by the National Natural Science Foundation of China under project (No. 62472126), Natural Science Foundation of Guangdong Province (Project No. 2023A1515011959), and Shenzhen-Hong Kong Jointly Funded Project (Category A, No. SGDX20230116 091246007).

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

# Appendix

## A Repo2Run details

### A.1 Example of Repo2Run

Figure 7 shows an example process of Repo2Run, including the event stream and Repo2Run workflow. Below, we introduce the executed action based on the example:

- **Base Image Change**: In the tenth part of Event Stream, when the LLM agent discovers that the "StrEnum" class cannot be imported through testing, which is only available in the Python standard library starting from version 3.11, it decides to reselect the base image to python:3.11.
- **Environment monitoring**: In the fourth part, the LLM agent calls the "`find`" command to search for `requirements.txt` within the environment to determine which dependencies to install.
- **Dependency installation**: In the sixth part, the action "`waitinglist addfile`" adds all elements from `requirements.txt` to the waiting list, while executing the "`download`" command. Third-party libraries are taken one by one from the waiting list and installed in the internal environment. The installation commands (e.g., "`pip install`" and "`apt-get install`") are executed.
- **Code editing**: The eighteenth part demonstrates code editing to correct a syntax error resulting from the improper use of double quotes in a Python f-string.
- **Bash commands**: The sixteenth part shows that the LLM agent uses the command "`export PYTHONPATH=/repo/src`".
- **Test running**: The eighth part shows that the LLM agent calls the "`runtest`" command to execute unit tests.

### A.2 Action usage frequency

As shown in Figure 8, we analyze the invocation times of various action types in 361 successfully built projects, including the five actions within the internal environment and the action of base image change. Bash commands are the most frequently used action, as they encompass the majority of instructions. Additionally, we observe that the LLM agent tends to call dependency installation quite frequently, averaging about 18 times per building, which means roughly 18 dependencies are installed per building on average. Moreover, the LLM agent calls test running approximately 3.5 times per building on average, which typically helps the agent better identify issues. Among successful buildings, we find 48 instances of changing the base image, accounting for 13.3% of success cases, indicating that the initial selection of the base image is often incorrect and requires subsequent adjustments.

## B Dockerfile synthesizer details

### B.1 Example

If the LLM agent successfully runs all tests in the repository in the internal environment, the building process needs to be recorded in a Dockerfile. Figure 9 illustrates the Dockerfile synthesizer's operation. Figure 9 (a) lists the internal commands actually executed in the internal environment, their return codes (e.g., 0 indicates success, otherwise failure), and the current directory (`dir`). Figure 9 (b) shows the process of generating the Dockerfile.

The principles for the conversion process include the following essential steps: scanning all internal commands sequentially, and the line numbers mentioned below precisely match the line number of Figure 9 (a):

1. By default, use `python:3.10` as the base image.
2. For commands that run successfully (i.e., have a return code of 0), prepend the command with "`RUN`" to form a Dockerfile statement (e.g., line 3). Commands that fail (i.e., have a return code other than 0, such as line 4) are rolled back and not included in the Dockerfile. An exception is that some commands typically do not change the current state (such as "`cat`" on line 2). They are not added to the Dockerfile. We show the complete list in Appendix B.2.

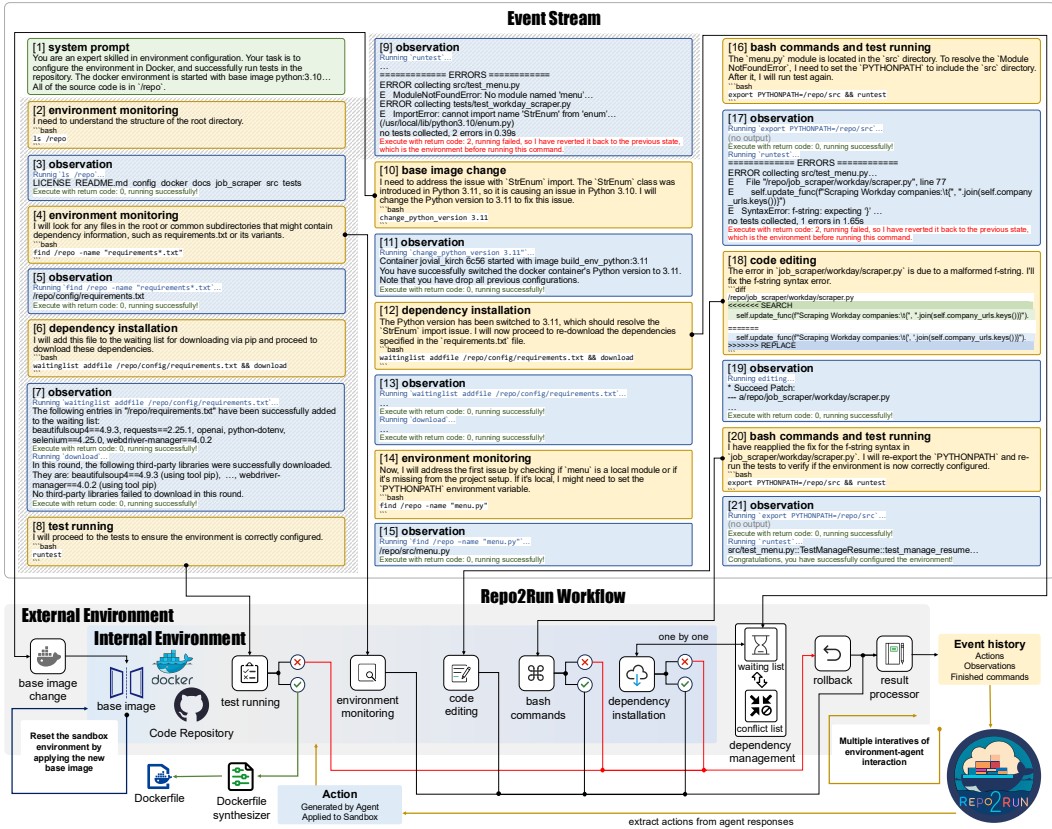

Figure 7: An example process of **Repo2Run**, which illustrates two main parts: **Event Stream** and **Repo2Run Workflow**. The Event Stream tracks the entire action-observation process, where the green box represents the abbreviated system prompt, yellow boxes represent responses from the LLM agent including the actions, and blue boxes represent observations from the Runtime Environment. The shaded actions indicate buildings abandoned after **base image change**. In the Event Stream, the blue text indicates that the command starts running, the green text indicates that the command runs successfully, and the red text indicates that the command fails. The **Repo2Run Workflow** consists of the **internal environment** and the **external environment**. The internal environment serves as the actual buildings Docker-based sandbox, which builds an actual testing runtime environment. The external environment executes the action-observation process and assists in the building process within the internal environment.

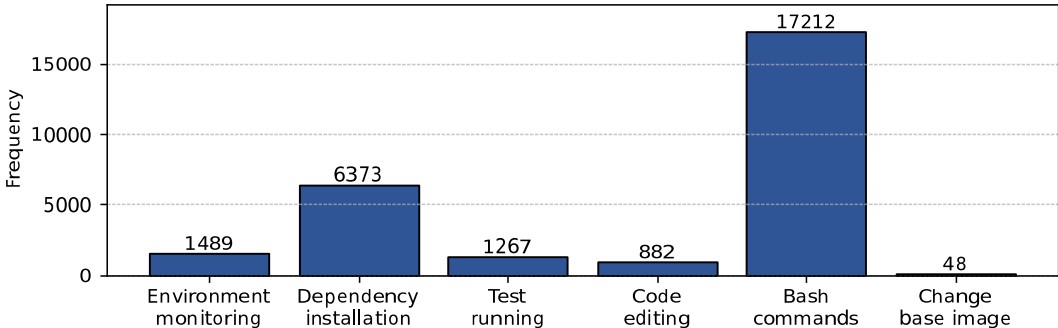

Figure 8: Action usage frequency of the success building.

| | internal command | return code | dir |
|---|---|---|---|
| 1 | ls /repo | 0 | / |
| 2 | cat /repo/README.md | 0 | / |
| 3 | pip install A | 0 | / |
| 4 | pip install -e /repo | 1 | / |
| 5 | change_python_version 3.11 | 0 | / |
| 6 | cd /repo | 0 | / |
| 7 | cat setup.py | 0 | /repo |
| 8 | pip install -e . | 0 | /repo |
| 9 | python code_edit.py 0.patch | 0 | /repo |
| 10 | poetry install | 1 | /repo |
| 11 | pip install B>=1.0, <2.0 | 0 | / |
| 12 | runtest | 1 | / |
| 13 | export TOKEN=123 | 0 | / |
| 14 | runtest | 0 | / |
| 15 | pipdeptree --json-tree | 0 | / |

(a) Commands running in the internal environment.

**0.** We begin with the default base image python:3.10.
```
FROM python:3.10
```

**1-4.** The commands "ls", "cat" are safe commands, so they won't be added into Dockerfile. Instead, we save the command "pip install A". The command "pip install -e ." failed, so rolling back.
```
FROM python:3.10
RUN pip install A
```

**5.** The base image is successfully changed, so all previous buildings are discarded, leaving only the new base image python:3.11.
```
FROM python:3.11
```

**6-8.** The commands "cd", "cat" are safe commands, so they will not be added into the Dockerfile. Instead, we save the command "cd /repo && pip install -e ."
```
FROM python:3.11
RUN cd /repo && pip install -e .
```

**9.** Successfully edited the file. Apply a patch in the external environment, and save the "python code_edit.py" statement.
```
FROM python:3.11
RUN cd /repo && pip install -e .
RUN cd /repo && python /code_edit.py /patch/0.patch
```

**10-12.** The command "poetry install" failed, so rolling back. Save the successful execution of "pip install B>=1.0, <2.0". Ignore all preset commands "runtest".
```
FROM python:3.11
RUN cd /repo && pip install -e .
RUN cd /repo && python /code_edit.py /patch/0.patch
RUN pip install B>=1.0, <2.0
```

**13.** The export statement is converted into an ENV command.
```
FROM python:3.11
RUN cd /repo && pip install -e .
RUN cd /repo && python /code_edit.py /patch/0.patch
RUN pip install B>=1.0, <2.0
ENV TOKEN=123
```

**14-15.** After passing the tests, use "pipdeptree" to record the actual installed versions of the current Python third-party libraries, and replace non-specific constraints in the Dockerfile with these actual versions. Additionally, since code editing is required, copy the "code_edit.py" script and patch file(s) into the Docker container.
```
FROM python:3.11
COPY code_edit.py /code_edit.py
COPY patch /patch
RUN cd /repo && pip install -e .
RUN cd /repo && python /code_edit.py /patch/0.patch
RUN pip install B==1.5.1
ENV TOKEN=123
```

(b) The process of synthesize the Dockerfile by Dockerfile synthesizer.

Figure 9: An example of Dockerfile synthesizer to transfer the commands into a runnable Dockerfile.

3. If `dir` is not the root directory, use `cd` to change to that directory before running the command, as each Dockerfile statement runs in its own directory session.

4. If a base image change statement is encountered (e.g., line 5), discard all previous buildings and switch to the new base image.

5. If a code editing command is encountered (e.g., line 9), copy the patch and editing script into the Docker container before executing the editing command.

6. If an export statement for adding environment variables is encountered (e.g., line 13), convert it to a persistent ENV statement.

7. After all statements have been scanned, check for dependency installation commands and replace any unspecified versions (e.g., "B>=1.0,<2.0" in line 11) with the actual versions downloaded in the Docker container (e.g., "B==1.5.1" in the final Dockerfile of Figure 9 (b)).

## B.2 Safe command

If Repo2Run executes the following commands without using ">" or ">>" for output redirection, they are regarded to be safe commands that typically do not affect the system. Therefore, rollback is not necessary, and they are not added to the generated Dockerfile.

["cd", "ls", "cat", "echo", "pwd", "whoami", "who", "date", "cal", "df", "du", "free", "uname", "uptime", "w", "ps", "pgrep", "top", "dmesg", "tail", "head", "grep", "find", "locate", "which", "file", "stat", "cmp", "diff", "xz", "unxz", "sort", "wc", "tr", "cut", "paste", "tee", "awk", "env", "printenv", "hostname", "ping", "traceroute", "ssh"]

## C  Pollution commands

Figure 10 shows two practical examples. As shown in Figure 10 (a), when we run "pip install cupy", the command fails to execute because the CUDA environment is invalid. However, it still introduces the "fastrlock" and "numpy" packages which is not present in the original environment. In Figure 10 (b), when we run the "rm -rf /path/to/logs" command to delete the directory, some files fail to be deleted due to a lack of permission. However, the files that have already been deleted (e.g., log1.txt and log2.txt) are not restored. Such "pollution" caused by failed commands may make the environment enter the uncertain state.

We also identified 30 Python packages that may introduce "pollution" when installing via pip, as listed in Table 6. This process can be easily reproduced in a clean Docker environment (such as the initial environment shown in Figure 10 (a)). For example, simply running "pip install adb" — even without specifying a version, which defaults to the latest — can cause such "pollution". This demonstrates that such scenarios are quite common in real-world development and also underscores the importance of the rollback action in Repo2Run.

```
user@hostname:/$ pip list
Package    Version
---------- -------
pip        23.0.1
setuptools 65.5.1
wheel      0.44.0
user@hostname:/$ pip install cupy
error: subprocess-exited-with-error
...
ERROR: Failed building wheel for cupy
Exception: Your CUDA environment is invalid.
user@hostname:/$ pip list
Package       Version
------------- -------
fastrlock     0.8.3
numpy         2.2.5
pip           23.0.1
setuptools    65.5.1
wheel         0.44.0
```

(a) Failed "pip install" example.

```
user@hostname:/path/to/logs$ tree /path/to/logs
/path/to/logs
├── log1.txt
├── log2.txt
└── secure
        ├── secure_log1.txt
        └── secure_log2.txt
user@hostname:/path/to/logs$ rm -rf /path/to/logs
rm: cannot remove '/path/to/logs/secure/secure_log1.txt':
Permission denied
rm: cannot remove '/path/to/logs/secure/secure_log2.txt':
Permission denied
rm: cannot remove '/path/to/logs/secure': Permission
denied
user@hostname:/path/to/logs$ tree /var/logs/app
/path/to/logs
└── secure
        ├── secure_log1.txt
        └── secure_log2.txt
```

(b) Failed "rm -rf" example.

Figure 10: Examples of failed command causing "pollution" to the environment.

Table 6: 30 examples of "pollution" during pip installation. "Package_name" represents the name of the package that failed to download, "pollution" represents other packages introduced by the failed download, and "pollution" count represents the number of these introduced packages.

| package_name | "pollution" | "pollution" count |
| --- | --- | --- |
| zbarlight | pillow | 1 |
| mxnet-cu91 | chardet, idna, urllib3 | 3 |
| texthero | blis, catalogue, certifi, charset-normalizer, click, . . . | 41 |
| url | six | 1 |
| robotframework-ride | certifi, charset-normalizer, idna, numpy, packaging, . . . | 10 |
| adb | libusb1, typing | 2 |
| onegov-core | fastcache, mailthon, passlib, polib, pytz, . . . | 12 |
| postal | six | 1 |
| changes | requests | 1 |
| mxnet-cu75mkl | chardet, idna, urllib3 | 3 |
| winpdb | numpy, six | 2 |
| slybot | attrs, Automat, certifi, chardet, charset-normalizer, . . . | 30 |
| fbprophet | aiohappyeyeballs, aiohttp, aiosignal, appdirs, async-timeout, . . . | 44 |
| mxnet-cu75 | chardet, idna, urllib3 | 3 |
| libarchive | nose | 1 |
| atari-py | numpy, six | 2 |
| reppy | cachetools, certifi, charset-normalizer, idna, python-dateutil, . . . | 8 |

Continued on next page

**Table 6 – continued from previous page**

| package_name | "pollution" | "pollution" count |
|---|---|---|
| sovrin | leveldb, libnacl, msgpack-python, orderedset, Pympler, ... | 15 |
| scrapely | numpy, six, w3lib | 3 |
| kevinsr | python-version | 1 |
| pysurvive | colored, numpy, pillow, psutil, pygtrie, ... | 6 |
| horovod | cffi, cloudpickle, packaging, psutil, PyYAML, ... | 6 |
| cupy | fastrlock, numpy | 2 |
| face-recognition | face-recognition-models | 1 |
| nsot | gunicorn, idna, ipaddress, ipython, itypes, ... | 15 |
| gooey | colored, numpy, pillow, psutil, pygtrie, ... | 6 |
| wxpython | numpy, six | 2 |
| neuralcoref | annotated-types, blis, boto3, botocore, catalogue, ... | 47 |
| apache-airflow-backport-providers-apache-hive | apispec, argcomplete, attrs, babel, cached-property, ... | 58 |
| bcolz | numpy | 1 |

## D   Benchmark statistic

As demonstrated in Table 7, we list the statistics on lines of code for repositories in our constructed benchmark.

Table 7: Statistics on Lines of Code (LOC) for repositories in our benchmark. The table below illustrates the statistics regarding the lines of code (LOC) of all repositories in the benchmark. It separately counts the number of repositories with LOC greater than 10,000 and greater than 100,000. The last two rows provide reference LOC for well-known industry-level repositories.

| Configuration state | #LOC > 10,000 | #LOC > 100,000 | Medium | Max |
|---|---|---|---|---|
| Success(361) | 220 (60.9%) | 27 (7.5%) | 14,578 | 6,431,084 |
| Fail(59) | 50 (84.7%) | 12 (30.6%) | 43,001 | 3,584,542 |
| All(420) | 270 (64.3%) | 39 (9.3%) | 15,854 | 6,431,084 |
| requests (reference) | Yes | No | 12,782 | 12,782 |
| pandas (reference) | Yes | Yes | 571,849 | 571,849 |

## E   Test pass rate experiment

We randomly sample 50 repositories from our benchmark, successfully run their entire test suites, and manually record the pass rates for each method. Table 8 shows the percentage of repositories for which each method achieved a test pass rate exceeding various thresholds (from >=10% to 100%).

Table 8: Pass rates across tools at varying thresholds. Each cell shows the percentage and the number of repositories in parentheses.

| Pass Rate Threshold | Repo2Run | SWE-agent | LLM generator | pipreqs |
|---|---|---|---|---|
| $\geq$10% | **70% (35)** | 26% (13) | 32% (16) | 10% (5) |
| $\geq$20% | **68% (34)** | 26% (13) | 32% (16) | 10% (5) |
| $\geq$30% | **64% (32)** | 26% (13) | 32% (16) | 10% (5) |
| $\geq$40% | **62% (31)** | 26% (13) | 32% (16) | 8% (4) |
| $\geq$50% | **62% (31)** | 24% (12) | 32% (16) | 8% (4) |
| $\geq$60% | **58% (29)** | 20% (10) | 30% (15) | 8% (4) |
| $\geq$70% | **54% (27)** | 18% (9) | 28% (14) | 8% (4) |
| $\geq$80% | **52% (26)** | 18% (9) | 26% (13) | 4% (2) |
| $\geq$90% | **48% (24)** | 18% (9) | 26% (13) | 4% (2) |
| 100% | **34% (17)** | 10% (5) | 14% (7) | 4% (2) |

The results are clear: regardless of the pass rate threshold chosen, Repo2Runconsistently and substantially outperforms all baselines. This experiment demonstrates that even when evaluated by the metric you suggested, our method's superiority remains clear.

We conduct this experiment to directly show that Repo2Runexcels not just in creating executable environments, but in creating high-quality, correct ones. We wish to briefly emphasize why we firmly believe that EBSR (Environment Building Success Rate) is an appropriate metric for this task.

Our goal is to automate the crucial first step of software usability: creating an executable environment. Test pass rates, while informative, are a "noisy" signal for environment quality because failures can arise from code bugs or test bugs, not just environment building errors. As industry practice (e.g., CI/CD) shows, distinguishing between a build/execution success and a test-pass success is a standard and necessary distinction.

# F    Empirical study

## F.1    Manual Experiment

### F.1.1    Settings

To ensure a fair comparison among participants, we conducted training and demonstrated examples before the experiment to ensure everyone understood the procedure. Additionally, to minimize discrepancies in time consumption due to network factors, all participants conducted the experiment in the same network environment.

### F.1.2    Survey

We selected eight technical staff from internet companies to participate in the experiment and conducted a survey regarding their backgrounds prior to the experiment. Their development experience ranges from 4 to 11 years, with an average of 7 years in software development and 3.8 years in Python development. Seven participants have experience in complex development projects, while one has experience in multiple small-scale projects.

Regarding environment building, three participants indicated that they spend a significant amount of time building the environment when faced with an unfamiliar code repository; another three stated that, although they spend a long time, it is generally manageable; two participants reported spending minimal time.

In terms of successful environment building in their regular work, three participants mentioned they only fail with extremely complex environments, while five indicated they can build environments for most medium-scale repositories. As for their confidence in successfully building unfamiliar environments, six participants expressed that they are usually successful, and two said they are sometimes successful.

When it comes to the amount of time they are willing to wait to build an unfamiliar repository, four participants are willing to wait for over 90 minutes, two are willing to wait 40-60 minutes, one is willing to wait 20-40 minutes, and one is only willing to wait 10-20 minutes.

Participants' overall evaluation of building code running environments is diverse: one finds it very troublesome with many issues, four find it somewhat troublesome, two indicate moderate difficulty, and one finds it relatively simple.

Additionally, all participants expressed a high or very high willingness to use a tool that could automatically build the environment for an unfamiliar repository.

### F.1.3 Experiment guideline

We request all participants to conduct the experiment following the guidelines below:

**1. Environment Setup**

Build the Docker environment. Verify the installation is successful: If installed correctly, you should be able to use the following command (python:3.10 is just an example; select the base image according to your requirements):

```
docker run -it python:3.10 bash
```

**2. Overall Procedure**

Our objective is to install the given package in a Docker container, build its environment, and be able to run its internal tests. During the process, record the time developers spend building the environment and eventually save the logs using docker logs.

**2.1 Review the Repository to be Built (Optional)**

Review the GitHub repository that needs to be built.

**2.2 Determine the Docker Base Image (Generally start with python:3.10)**

Select the Docker base image based on the repository (all repositories use Python as the main language). Common base images include: Note, in this experiment, it is generally sufficient to use the official Python series images. The main concern is the version; if uncertain, you can start with newer versions like 3.10, 3.11, 3.12, or select the recommended Python version based on the repository's README.

Python series (the number after python indicates the version): python:3.10, python:3.12, python:3.9, python:3.6, python:3.9... PyTorch series: pytorch/pytorch, pytorch/pytorch:1.9.0-cuda10.2-cudnn7-runtime... Anaconda series: continuumio/anaconda3... Note: If you find the selected version is incorrect later, you can exit and reselect.

**2.3 Create and Enter the Container**

Using the determined base image name (e.g., python:3.10), enter the container with the following command:

```
docker run -it -name mytest python:3.10 bash
```

Here, mytest is the container name, recorded for log export later. It can be freely named, just keep track of it to avoid losing it later. Note: If any issues arise here, check if Docker is correctly installed and if the image name is valid, and troubleshoot accordingly.

**2.4 Install Relevant Tools**

APT tool downloads:

```
apt-get update && apt-get install -y curl
```

Download pytest:

```
pip install pytest
```

**2.5 Download the Repository**

Select the repository to be built and download its GitHub repository to a location (generally directly in the root directory):

```
git clone https://github.com/{full_name}.git
```

## 2.6 Build the Environment

Now, use your skills to build: First, enter the downloaded file directory, for example:

```
cd wddbfs
```

Switch to the specified branch SHA (refer to the corresponding SHA of the repository), for example:

```
git checkout 5c68aa
```

Our goal is to successfully run pytest (not necessarily to pass all tests, just to run them). A simple criterion is to successfully run:

```
pytest -collect-only -q
```

At this point, you can use your experience and information from the repository documentation and debugging error messages to build. However, there are a few restrictions:

Do not directly edit the test files! (Files starting with test_ or ending with _test.py). Do not directly delete test files! Editing the original repository files is not recommended.

During this process, you may perform various operations, including but not limited to pip, apt-get, and other tool downloads, as well as searching online or using GPT for debugging help.

Additionally, if there are long download times requiring waiting, you may decide according to your situation whether to leave this running and do other things (just don't forget about this task).

## 2.7 Completion and Logging

A task can conclude in two scenarios:

Scenario One: If "`pytest -collect-only -q`" runs without issues, you can then execute pytest. If pytest completes successfully, the task is done. Scenario Two: If you feel the package is extremely difficult to build, for example exceeding your patience threshold (refer to your usual development habits), you may also terminate.

Once finished, input exit to exit. Make sure to save your output logs with the following command (replace container_name with the container name you recorded earlier, if you forget, you can use docker ps to check):

```
docker logs container_name -t > wddbfs.log
```

## 2.8 Fill Out the Form and Record Information

You need to fill out the form according to your feelings.

Here, Table 9 presents the form that developers are required to complete after the environment building process, including items such as "Is it successful?", "Final base image used", "Reason for failure", and so on. Besides, Tables 10 and 11 show two examples filled out by developers.

Table 9: The form that developers should fill in after the environment building.

| Question | Description |
| --- | --- |
| Is it successful? | Yes or No - whether the task successfully passed "`pytest -collect-only -q`" without errors and eventually ran "`pytest`". |
| Final base image used | E.g., python:3.10 |

Continued on next page

| Question | Description |
|---|---|
| Reason for failure | Summarize the main reasons for failure (if failed), including:
• Long download time
• Difficulty handling repository dependencies
• Unresolvable bug
• Errors in repository tests
• Lengthy test durations
• Other reasons (please specify) |
| Waiting time | Approximate value, indicating the time spent waiting for dependencies to download. Does not include time spent on decision-making and research. Provide an approximate range:
• <3 minutes
• 3-5 minutes
• 6-10 minutes
• 11-20 minutes
• 20-40 minutes
• 40-60 minutes
• 60-90 minutes
• 90+ minutes |
| Longest time-consuming process | Describe the most time-consuming building process, such as:
• Downloading a specific dependency
• Resolving a specific error
• Incorrect Python version selection |
| Tolerance level | Based on your subjective feeling and development experience. Rate the process (1-5):
• 1: Extremely unbearable
• 2: Somewhat unbearable, but manageable
• 3: Neutral, tolerable
• 4: Comfortable, no significant discomfort
• 5: Very comfortable, highly satisfactory building experience |
| Building difficulty | Based on the complexity of the building process (ignoring time spent): Rate the process (1-5):
• 1: Very simple, completed with intuition and experience, no reference materials needed
• 2: Fairly simple, referred to basic materials (e.g., README), simple overall steps
• 3: Moderate difficulty, encountered some issues, but manageable
• 4: Difficult, required extensive debugging and building
• 5: Very difficult, needed numerous references and encountered unresolved or time-consuming issues |
| Materials referenced | List the materials used for reference (e.g., README, internal building files, online searches, GPT). If none, mention "Directory". |
| Biggest challenge during the process | Describe the most troublesome aspect of the building process, such as long wait times, unclear error messages, dependency version conflicts, inability to find required software versions, etc. |

**Example 1**:

Table 10: A sample form filled by a developer after environment building, illustrating a successful environment building.

| Question | Answer |
|---|---|
| Is it successful? | Yes |
| Final base image used | python:3.10 |
| Reason for failure | Successful |
| Waiting time | 3-5 minutes |
| Longest time-consuming process | Downloading poetry |
| Tolerance level | 5 |
| Building difficulty | 1 |
| Materials referenced | Directory |
| Biggest challenge during the process | Waiting for building and installation |

**Example 2**:

Table 11: A sample form filled by a developer after environment building, illustrating a failed environment building.

| Question | Answer |
|---|---|
| Is it successful? | No |
| Final base image used | python:3.11 |
| Reason for failure | Unresolvable bug, provide bug image |
| Waiting time | 40-60 minutes |
| Longest time-consuming process | Resolving a series of FileNotFoundError, ImportError, incorrect Python version selection |
| Tolerance level | 2 |
| Building difficulty | 5 |
| Materials referenced | README, GPT, StackOverflow |
| Biggest challenge during the process | Dependency version conflicts, long wait times |

### F.1.4 Repository assignment

We randomly assigned each participant four unique code repositories that were successfully built by Repo2Run. Additionally, each participant was assigned two code repositories that Repo2Run failed to build. To avoid chance occurrences, each failed repository was assigned to two different participants. Below is the list of selections:

**Successfully built**:

[alexwlchan/safari-webarchiver, ManiMozaffar/aioclock, mixedbread-ai/batched, mobiusml/gemlite, circlemind-ai/fast-graphrag, knowsuchagency/promptic, mbodiai/embodied-agents, modelscope/agentscope, Adibvafa/CodonTransformer, kennethreitz/simplemind, lmstudio-ai/venvstacks, mlecauchois/micrograd-cuda, IST-DASLab/PanzaMail, MetaGLM/zhipuai-sdk-python-v4, openai/mle-bench, RealOrangeOne/django-tasks, basf/MolPipeline, dai-motoki/zoltraak, lucidrains/alphafold3-pytorch, mistralai/mistral-common, BMPixel/moffee, DataformerAI/-dataformer, jahwag/ClaudeSync, volfpeter/htmy, Genentech/gReLU, OpenNLPLab/lightning-attention, paradigmxyz/spice, reagento/dishka, arcee-ai/fastmlx, KyanChen/RSMamba, neuralmagic/guidellm, simonw/files-to-prompt]

**Failed to build**:

[zhuqinfeng1999/Samba, dongxuyue/Open-ReplaceAnything, LazyAGI/LazyLLM, jialuechen/deep-folio, KOSASIH/pi-nexus-autonomous-banking-network, AARG-FAN/Yolo_for_Wukong, plinder-org/plinder, expectedparrot/edsl]

### F.1.5 Results

Based on the times shown in the logs, we calculated that the average building time for each repository is 21.33 minutes. Furthermore, none of the repositories that Repo2Run failed to build were successfully built manually. Additionally, in the manual experiment, five environments that were successfully built by Repo2Run were not successfully built, representing 15.6% of the total successfully built assignments.

The table information and logs are too lengthy, so we place them on `https://anonymous.4open.science/r/Repo2Run`.

### F.2 Industry-level repository building validation

As shown in Table 12, we present the evaluation of Repo2Run on 59 popular industry-grade repositories. Additionally, Table 13 shows the successful configuration of additional industry-grade repositories.

Table 12: Evaluation of Repo2Run on 59 popular industry-grade repositories. The repositories were selected from GitHub Ranking [60], filtering out those without tests and with #LOC < 10,000.

| Configuration state | #LOC > 10,000 | #LOC > 100,000 | Medium | Max |
|---|---|---|---|---|
| Success(30) | 30 (100%) | 14 (46.7%) | 96,157 | 2,389,065 |
| Fail(29) | 29 (100%) | 24 (82.6%) | 333,464 | 2,669,973 |
| All(59) | 59 (100%) | 40 (9.3%) | 124,352 | 2,669,973 |

Table 13: Successful additional configuration of 30 industry-grade repositories.

| Repository | #LOC | Repository | #LOC | Repository | #LOC |
|---|---|---|---|---|---|
| comfyanonymous/ ComfyUI | 737,543 | OpenBB-finance/OpenBB | 2,389,065 | scrapy/scrapy | 77,278 |
| yt-dlp/yt-dlp | 155,662 | keras-team/keras | 204,537 | AUTOMATIC1111/ stable-diffusion-webui | 40,448 |
| pallets/flask | 25,190 | geekan/MetaGPT | 82,305 | psf/black | 118,551 |
| deepfakes/faceswap | 74,690 | streamlit/streamlit | 201,894 | psf/requests | 12,782 |
| lllyasviel/Fooocus | 328,199 | labmlai/annotated_ deep_learning _pa-per_implementations | 510,851 | RVC-Boss/GPT-SoVITS | 34,658 |
| mingrammer/ dia-grams | 11,994 | OpenInterpreter/open-interpreter | 15,818 | hiyouga/LLaMA-Factory | 76,000 |
| nvbn/thefuck | 10,267 | pandas-dev/pandas | 571,849 | ytdl-org/youtube-dl | 96,157 |
| TheAlgorithms/ Python | 419,917 | fastapi/fastapi | 177,120 | Textualize/rich | 45,744 |
| All-Hands-AI/OpenHands | 160,514 | Stability-AI/ stablediffusion | 13,698 | ultralytics/ultralytics | 94,201 |
| gradio-app/gradio | 184,722 | QuivrHQ/quivr | 10,264 | freqtrade/freqtrade | 188,889 |

## G Repo2Run tools

Showing in Table 14, we design the following actions for Repo2Run to facilitate its invocation.

Table 14: Command list and their functions.

| Command | Function |
| --- | --- |
| `waitinglist add -p package_name [-v version_constraints] -t tool` | Add item into waiting list. If no "version_constraints" are specified, the latest version will be downloaded by default. |
| `waitinglist addfile file_path` | Add all entries from a file similar to requirements.txt format to the waiting list. Format should be package_name [version_constraints]. |
| `waitinglist clear` | Clear all items in the waiting list. |
| `conflictlist solve -v "[version_constraints]"` | Resolve the conflict for the first element in the conflict list, and update the version constraints for the corresponding package_name and tool to version_constraints. If no "version_constraints" are specified, the latest version will be downloaded by default. The package_name and tool in the original waiting list must match one of the elements in the conflictlist. Here, the version_constraints are specified. |
| `conflictlist solve -u` | Keep the original version constraint that exists in the waiting list, and discard the other version constraints with the same name and tool in the conflict list. |
| `conflictlist clear` | Clear all items in the conflict list. |
| `conflictlist show` | Show all items in the conflict list. |
| `waitinglist show` | Show all items in the waiting list. |
| `download` | Download all pending items in the waiting list at once, and the conflict list must be empty before executing. |
| `runtest` | Check if the built environment is correct using "pytest". |
| `poetryruntest` | Check if the built environment is correct in the poetry environment. If you want to run tests in the poetry environment, run it. |
| `runpipreqs` | Generate requirements_pipreqs.txt and pipreqs_output.txt and pipreqs_error.txt. |
| `change_python_version python_version` | Switching the Python version in the Docker container will forgo any installations made prior to the switch. The Python version number should be represented directly with numbers and dots, without any quotation marks. |
| `clear_configuration` | Reset all the buildings to the initial setting of `python:3.10`. |

## H  Repository building success status

As shown in Table 15, we list the success status of all the packages in our constructed benchmark.

Table 15: Success status of each package in the benchmark.

| full_name | sha | success | full_name | sha | success |
|---|---|---|---|---|---|
| 271374667/VideoFusion | 9ba7b8 | Yes | 6abd/horus | c1d093 | Yes |
| a-r-r-o-w/cogvideox-factory | 80d115 | Yes | a-s-g93/neo4j-runway | 16b441 | Yes |
| Aaditya-Prasad/consistency-policy | eed0c4 | No | AARG-FAN/Yolo_for_Wukong | 07f61a | No |
| adamobeng/wddbfs | 5c68aa | Yes | Adibvafa/CodonTransformer | 2842ef | Yes |
| AdityaNG/kan-gpt | 0c6e4c | Yes | Admyral-Security/admyral | de332e | Yes |
| AgentOps-AI/AgentStack | ff9c6a | Yes | aidatatools/ollama-benchmark | c6a5fd | Yes |
| AIR-Bench/AIR-Bench | 4b27b8 | Yes | airbytehq/PyAirbyte | 7e65ab | Yes |
| airtai/fastagency | 1ff503 | Yes | Akkudoktor-EOS/EOS | fff685 | Yes |
| alexmolas/microsearch | 632ff2 | Yes | alexwlchan/safari-webarchiver | 0e4974 | Yes |
| AlibabaPAI/llumnix | b319b2 | Yes | All-Hands-AI/OpenHands | 246107 | Yes |
| alvin-r/databonsai | 3f2b7c | Yes | amchii/tg-signer | 926dbb | Yes |
| andrewyng/aisuite | 763996 | Yes | andrewyng/translation-agent | e0fc60 | Yes |
| AnswerDotAI/byaldi | 4583c0 | Yes | AnswerDotAI/rerankers | ecd1f6 | Yes |
| antgroup/agentUniverse | ed8f55 | Yes | apapiu/transformer_latent_diffusion | 84a75e | Yes |
| apify/crawlee-python | 267063 | Yes | apple/ToolSandbox | 1a1dc8 | Yes |
| apple/ml-cross-entropy | 1f3ebd | Yes | apple/ml-mdm | 9a5632 | Yes |
| arcee-ai/fastmlx | fd37bc | Yes | argmaxinc/whisperkittools | 03898f | Yes |
| arvindrajan92/DTrOCR | a10aa0 | Yes | astramind-ai/Auralis | c357a1 | Yes |
| atonderski/neuro-ncap | ecdcf2 | Yes | aurelio-labs/semantic-chunkers | 04acc2 | Yes |
| AuvaLab/itext2kg | 941a1d | Yes | awslabs/agent-evaluation | 3df695 | Yes |
| Azure/co-op-translator | a4709e | Yes | Azure-Samples/rag-postgres-openai-python | 61bde7 | Yes |
| bananaml/fructose | 5f24ec | Yes | basf/MolPipeline | 2f9bae | Yes |
| basf/mamba-tabular | af1ea0 | Yes | beatzxbt/mm-toolbox | 728e35 | Yes |
| bellingcat/ShadowFinder | 578d5a | Yes | Benexl/FastAnime | 677f46 | Yes |
| betaacid/FastAPI-Reference-App | 8caeca | Yes | bhavnicksm/chonkie | 990493 | Yes |
| bigcode-project/bigcodebench | aa634d | Yes | Bl3f/yato | 4906b0 | Yes |
| Blealtan/efficient-kan | 7b6ce1 | Yes | block/goose | c497a5 | Yes |
| BMPixel/moffee | 0e643d | Yes | boheumd/MA-LMM | ffe9fa | Yes |
| bytewiz3/TravelGPT | b19b43 | Yes | CausalLearning/ReAct | 7d3665 | No |
| cfahlgren1/observers | d46fdb | Yes | chaidiscovery/chai-lab | b6e7fa | Yes |
| character-ai/prompt-poet | 466432 | Yes | cheahjs/palworld-save-tools | 7dc2c7 | Yes |
| chernyadev/bigym | 72d305 | Yes | chrschy/fact-finder | ca57d1 | Yes |
| circlemind-ai/fast-graphrag | 447511 | Yes | cloudflare/cloudflare-python | 228479 | Yes |
| codefuse-ai/CodeFuse-muAgent | e93924 | No | codeintegrity-ai/mutahunter | f88922 | Yes |
| codematrixer/hmdriver2 | c0d075 | Yes | codeskyblue/tidevice3 | d83c34 | Yes |
| codeskyblue/uiautodev | eb8577 | Yes | Codium-ai/AlphaCodium | eb7577 | Yes |
| Codium-ai/cover-agent | 5c4b88 | Yes | COLA-Laboratory/TransOPT | de8bf3 | No |
| Comfy-Org/comfy-cli | 7711db | Yes | CompEpigen/figeno | 14b904 | Yes |
| computer-agents/agent-studio | d7f6cb | Yes | cosmic-cortex/mlfz | 5bf8d2 | Yes |
| cremebrule/digital-cousins | 49400b | Yes | crewAIInc/crewAI-tools | 873935 | Yes |
| cvg/nerf-on-the-go | 3659e7 | Yes | D-Star-AI/dsRAG | 2d5431 | Yes |
| D4Vinci/Scrapling | 012820 | Yes | DAGWorks-Inc/burr | 79137e | Yes |
| dai-motoki/zoltraak | 4dce44 | Yes | darrenburns/posting | 94feab | Yes |
| DataformerAI/dataformer | 0cf88c | Yes | daxa-ai/pebblo | e67b01 | Yes |
| daya0576/beaverhabits | c01257 | Yes | dbos-inc/dbos-transact-py | d6c6ac | Yes |
| deepsense-ai/db-ally | 26033f | Yes | dendrite-systems/dendrite-python-sdk | 27c9da | Yes |
| denser-org/denser-retriever | 76256e | No | dingo-actual/infini-transformer | 08d0a1 | Yes |
| discord/access | 19e9b1 | Yes | dleemiller/WordLlama | e38d47 | Yes |
| dongxuyue/Open-ReplaceAnything | 83f0ae | No | dottxt-ai/outlines-core | 31ab9f | Yes |
| dottxt-ai/prompts | 3d2689 | Yes | dreadnode/rigging | 82ac80 | Yes |
| droid-dataset/droid_policy_learning | 205ff6 | Yes | DS4SD/docling | aee9c0 | Yes |
| dynamiq-ai/dynamiq | 6cca1c | Yes | eakmanrq/sqlframe | 61fda5 | Yes |
| EleutherAI/sae | 0483b5 | Yes | emcf/thepipe | 02e397 | Yes |

Continued on next page

| full_name | sha | success | full_name | sha | success |
|---|---|---|---|---|---|
| Emerging-AI/ENOVA | b3661d | Yes | EnhancedJax/Bagels | d72d7f | Yes |
| enoch3712/ExtractThinker | 4872a7 | No | epic-open-source/seismometer | b3e812 | Yes |
| epistoteles/TensorHue | 1564fa | Yes | epogrebnyak/justpath | 0aca51 | Yes |
| erezsh/reladiff | d8683b | Yes | etianen/logot | 54e5ef | Yes |
| expectedparrot/edsl | aa7a2d | No | explosion/spacy-layout | 64c6f4 | Yes |
| facebookresearch/audioseal | ea10f5 | Yes | facebookresearch/lightplane | 34fe6c | Yes |
| facebookresearch/spiritlm | 52fb2f | Yes | FalkorDB/GraphRAG-SDK | 250ebe | Yes |
| Fanqi-Lin/Data-Scaling-Laws | bd6941 | No | fastapi/fastapi-cli | bc0840 | Yes |
| fedirz/faster-whisper-server | cbb6c9 | Yes | felafax/felafax | 34a475 | Yes |
| filipstrand/mflux | 627398 | Yes | FlagOpen/FlagGems | ca13b7 | No |
| fmind/cookiecutter-mlops-package | 00fef7 | Yes | foundation-model-stack/fms-fsdp | 408c75 | Yes |
| fpgmaas/cookiecutter-uv | 90de47 | Yes | frdel/agent-zero | 3cefa1 | Yes |
| frostming/tetos | 106ea5 | Yes | Fugaku-LLM/DeepSpeedFugaku | 74753f | No |
| gauge-sh/bridge | 8b3430 | Yes | Genentech/gReLU | efd308 | Yes |
| genomoncology/FuzzTypes | d96243 | Yes | getludic/ludic | a6db96 | Yes |
| getzep/graphiti | 9f3dd5 | Yes | GigaxGames/gigax | c3c209 | Yes |
| gojasper/flash-diffusion | 48e3bc | Yes | gomate-community/TrustRAG | 1334c4 | Yes |
| gomate-community/rageval | 01e258 | Yes | google-deepmind/nanodo | 10aefd | Yes |
| google-deepmind/penzai | fda6cd | Yes | google-deepmind/treescope | dac18b | Yes |
| google-research/timesfm | 02bc2f | Yes | goombalab/hydra | b6b9b7 | Yes |
| gpustack/gpustack | 4f0c67 | Yes | gregpr07/browser-use | 5e545d | Yes |
| groq/groq-python | fa2e13 | Yes | gusye1234/nano-graphrag | 18fa3a | Yes |
| hailo-ai/hailo-rpi5-examples | 82cfc8 | Yes | Haiyang-W/GiT | ef2b64 | No |
| HanaokaYuzu/Gemini-API | e8a2d2 | Yes | HATTER-LONG/Verbiverse | 82f988 | Yes |
| hinthornw/trustcall | eaaaad | Yes | HKUDS/HiGPT | 2b0793 | No |
| HKUDS/UrbanGPT | be3515 | No | hngprojects/ hng_boilerplate_python_fastapi_web | bc9740 | Yes |
| hpcaitech/Open-Sora | 38de63 | Yes | hpcaitech/SwiftInfer | 239fd3 | No |
| hrnoh24/stream-vc | faa629 | Yes | huchenlei/ComfyUI_omost | 7ef00d | Yes |
| huggingface/lerobot | 4c41f6 | Yes | huggingface/lighteval | 6ad727 | Yes |
| HZAI-ZJNU/Mamba-YOLO | ea97fc | No | IAAR-Shanghai/Grimoire | 3fe89d | Yes |
| ib-api-reloaded/ib_async | 38cf54 | Yes | IBM/fastfit | 396611 | Yes |
| IBM/terratorch | 16e5af | Yes | IEIT-Yuan/Yuan2.0-M32 | b403a2 | No |
| igorbenav/SQLModel-boilerplate | 2ead04 | Yes | igorbenav/fastcrud | dc831b | Yes |
| igrek51/wat | 0d6079 | Yes | illuin-tech/colpali | e45c4c | Yes |
| illuin-tech/vidore-benchmark | 469665 | Yes | Indoxer/LKAN | 16c48e | No |
| Infini-AI-Lab/Sequoia | 688079 | No | instanseg/instanseg | 0df8b2 | Yes |
| instructlab/instructlab | c978b2 | Yes | Integuru-AI/Integuru | 928e82 | Yes |
| InternLM/InternEvo | 5ad2eb | Yes | invariantlabs-ai/invariant | 81547a | Yes |
| IST-DASLab/PanzaMail | b1807c | Yes | iterative/datachain | b67d59 | Yes |
| IvanDrokin/torch-conv-kan | 7a0e83 | Yes | jahwag/ClaudeSync | 000633 | Yes |
| jgravelle/pocketgroq | e995c4 | Yes | jhj0517/AdvancedLivePortrait-WebUI | a7975c | Yes |
| jialuechen/deepfolio | 15d247 | No | jina-ai/late-chunking | db558c | Yes |
| jlowin/fastmcp | baa300 | Yes | jmschrei/tangermeme | a96897 | Yes |
| jonbarron/camp_zipnerf | 8e6d57 | Yes | JosephBARBIERDARNAL/pypalettes | 826930 | Yes |
| JoshuaC215/agent-service-toolkit | c72f48 | Yes | jshuadvd/LongRoPE | eb9aba | Yes |
| jxnl/n-levels-of-rag | 2ce110 | Yes | karpathy/minbpe | 1acefe | Yes |
| kennethreitz/simplemind | 39b5a5 | Yes | kevinzakka/mink | cf1a30 | Yes |
| knowsuchagency/promptic | a1930c | Yes | koaning/uvtrick | 2d7f27 | Yes |
| kohjingyu/search-agents | 7c35ac | No | KOSASIH/pi-nexus-autonomous-banking-network | 7fcff4 | No |
| kotaro-kinoshita/yomitoku | 71c85b | Yes | KruxAI/ragbuilder | db3d3d | No |
| kujirahand/tkeasygui-python | b1f293 | No | KyanChen/RSMamba | 3fa305 | Yes |
| kyegomez/MultiModalMamba | 58db40 | No | landing-ai/vision-agent | 63eab8 | Yes |
| langchain-ai/langchain-postgres | 064e5b | Yes | lavague-ai/LaVague | b3557f | Yes |
| LazyAGI/LazyLLM | e0dd38 | No | lenML/Speech-AI-Forge | 0b31b2 | Yes |
| leopiney/neuralnoise | c0313f | Yes | lichao-sun/Mora | 7a030e | No |
| Lightning-AI/litdata | 0a97de | Yes | lightonai/pylate | 8de184 | No |
| LilianHollard/LeYOLO | 872841 | Yes | line/lighthouse | ba9da7 | Yes |
| LlmKira/fast-langdetect | 5728ba | Yes | LMCache/LMCache | 7d3443 | Yes |

| full_name | sha | successful | full_name | sha | success |
|---|---|---|---|---|---|
| lmstudio-ai/mlx-engine | daeb7a | Yes | lmstudio-ai/venvstacks | 235ce3 | Yes |
| lucasdelimanogueira/PyNorch | ed391e | Yes | LucasFaudman/apkscan | 3b3e62 | Yes |
| lucidrains/alphafold3-pytorch | 49f7c9 | Yes | lucidrains/infini-transformer-pytorch | 5774bb | Yes |
| lucidrains/pi-zero-pytorch | 8ad66f | Yes | lucidrains/titok-pytorch | 2f9525 | Yes |
| lucidrains/transfusion-pytorch | 16f73e | Yes | MadcowD/ell | 36ca5e | Yes |
| ManiMozaffar/aioclock | 3d196b | Yes | MaoXiaoYuZ/Long-Novel-GPT | e952ac | No |
| Marker-Inc-Korea/AutoRAG | aa0bfb | Yes | martius-lab/hitchhiking-rotations | 45b49f | Yes |
| mbodiai/embodied-agents | 8715f6 | Yes | McGill-NLP/weblinx | 6f2014 | Yes |
| McGill-NLP/webllama | 696a7c | Yes | Menghuan1918/pdfdeal | e08199 | Yes |
| meta-llama/llama-stack-apps | f14a73 | Yes | MetaGLM/zhipuai-sdk-python-v4 | 7ff4de | Yes |
| metavoiceio/metavoice-src | de3fa2 | Yes | microsoft/MInference | 7a3e5a | No |
| microsoft/TinyTroupe | 9b8d4e | Yes | microsoft/Trace | 826cf5 | Yes |
| microsoft/aurora | 8b1165 | Yes | microsoft/graphrag | de1252 | Yes |
| microsoft/semantic-link-labs | 8e37ef | Yes | mikekelly/AgentK | e9ec89 | Yes |
| Mindinventory/MindSQL | 3d0ff0 | Yes | MinishLab/model2vec | 4e3fba | Yes |
| miquido/draive | 270f0c | Yes | mistralai/mistral-common | 5cac5e | Yes |
| mistralai/mistral-finetune | 656df1 | Yes | mixedbread-ai/baguetter | a6e915 | Yes |
| mixedbread-ai/batched | 1a1797 | Yes | mkjt2/lockbox | 58430d | Yes |
| mlecauchois/micrograd-cuda | ab1ca0 | Yes | MLT-OSS/open-assistant-api | 44eeaf | Yes |
| mlx-graphs/mlx-graphs | 4619d9 | No | mobiusml/gemlite | 5ebcca | Yes |
| ModelCloud/GPTQModel | a5aefc | No | modelcontextprotocol/python-sdk | aaf32b | Yes |
| modelscope/MemoryScope | 330b76 | Yes | modelscope/agentscope | ceaf89 | Yes |
| modern-python/that-depends | 65e656 | Yes | muchdogesec/history4feed | 614182 | Yes |
| muditbhargava66/PyxLSTM | f3c9bb | Yes | narwhals-dev/narwhals | a2088f | Yes |
| nasa-jpl/rosa | 5471dc | Yes | neo4j/neo4j-graphrag-python | 0ac06b | Yes |
| neuralmagic/AutoFP8 | e94461 | Yes | neuralmagic/guidellm | ecf298 | Yes |
| NewT123-WM/tnlearn | 50ee75 | Yes | NexaAI/nexa-sdk | 33f6ba | No |
| nicobrenner/commandjobs | 4c7264 | Yes | Nike-Inc/koheesio | 9bd29e | Yes |
| nlmatics/nlm-ingestor | c72542 | Yes | NLPJCL/RAG-Retrieval | d73057 | No |
| nomic-ai/contrastors | 496a05 | No | NousResearch/finetuning-subnet | e2f5eb | Yes |
| NUS-HPC-AI-Lab/VideoSys | 6c92ae | No | NVIDIA/Megatron-Energon | 28aa3b | Yes |
| NVIDIA/NeMo-Skills | 5591f3 | Yes | NVIDIA/kvpress | 715f8a | Yes |
| NVIDIA/logits-processor-zoo | db179a | Yes | NVIDIA/nv-ingest | eec9fa | No |
| NVlabs/Sana | 41dcbe | Yes | NVlabs/VILA | ec7fb2 | No |
| NVlabs/nvTorchCam | cc27be | Yes | ogkalu2/comic-translate | 1933d1 | Yes |
| Open-Wine-Components/umu-launcher | b0c0d4 | Yes | openai/mle-bench | 51ec2b | Yes |
| openai/swarm | 9db581 | Yes | opendatalab/MinerU | 391a99 | Yes |
| openfoundry-ai/model_manager | 34f9ff | Yes | opengeos/HyperCoast | c1604c | Yes |
| OpenInterpreter/aifs | 3f74c6 | Yes | OpenNLPLab/lightning-attention | d74395 | Yes |
| openpsi-project/ReaLHF | 62d9cd | Yes | openrecall/openrecall | 225a27 | Yes |
| OpenSPG/KAG | 68b2c6 | No | orbital-materials/orb-models | 251573 | Yes |
| outspeed-ai/outspeed | 049b40 | Yes | OwlAIProject/Owl | 919226 | Yes |
| PacktPublishing/LLM-Engineers-Handbook | ec6717 | Yes | paradigmxyz/spice | e962a9 | Yes |
| patched-codes/patchwork | c9b02b | Yes | paulrobello/parllama | 421238 | Yes |
| PeiJieSun/NESCL | 365d20 | Yes | plinder-org/plinder | 9658cc | No |
| pomonam/kronfluence | 884255 | Yes | PrefectHQ/ControlFlow | f259fa | Yes |
| PrimeIntellect-ai/OpenDiloco | 71f5c2 | Yes | PrimeIntellect-ai/prime | a974cf | Yes |
| princeton-nlp/SWE-agent | 8b3571 | Yes | proger/accelerated-scan | db7145 | Yes |
| pydantic/logfire | 3d7924 | Yes | pymupdf/RAG | b25718 | No |
| pytorch-labs/LeanRL | a416e6 | Yes | raphaelmansuy/code2prompt | 3b377b | No |
| RapidAI/RapidDoc | 5e5fef | Yes | RapidAI/RapidLayout | 8e9677 | Yes |
| reagento/dishka | 2ed985 | Yes | real-stanford/universal_manipulation_interface | 298776 | No |
| Realiserad/fish-ai | f32c7f | Yes | RealOrangeOne/django-tasks | e6d26c | Yes |
| reidjs/text-scheduler | 8bb7d6 | Yes | reka-ai/reka-vibe-eval | 93ecd9 | Yes |
| remigenet/TKAN | 8a1de0 | Yes | rio-labs/rio | eda40a | Yes |
| robocasa/robocasa | 27f992 | Yes | RobotecAI/rai | d15910 | No |
| robusta-dev/holmesgpt | c4743a | Yes | royreznik/rexi | f1dca8 | Yes |

| full_name | sha | success | full_name | sha | success |
|---|---|---|---|---|---|
| run-llama/llama_deploy | 47efff | Yes | run-llama/llama_extract | 89438f | Yes |
| run-llama/llama_parse | f78186 | Yes | SamKhoze/ComfyUI-DeepFuze | edd7fe | No |
| ScrapeGraphAI/Scrapegraph-ai | bae92b | Yes | seanchatmangpt/dspygen | 69f305 | No |
| serverless-ca/terraform-aws-ca | 2da837 | No | ServerlessLLM/ServerlessLLM | 8f1e6b | Yes |
| ServiceNow/BrowserGym | 12aa5e | Yes | ServiceNow/TapeAgents | 3eca5c | Yes |
| ServiceNow/WorkArena | 0ab9cb | No | ShaShekhar/aaiela | 4e8d6a | No |
| shawntan/scattermoe | 63b76a | Yes | ShoggothAI/motleycrew | 19837e | Yes |
| showlab/computer_use_ootb | 419d9d | Yes | shun-liang/yt2doc | 201ec2 | Yes |
| siliconflow/BizyAir | cdb3bb | Yes | simonw/files-to-prompt | f9a4d8 | Yes |
| simonw/llm-claude-3 | c62bf2 | Yes | simonw/llm-cmd | 74fb98 | Yes |
| simonw/llm-jq | beaada | Yes | simular-ai/Agent-S | ca83be | Yes |
| sirocco-ventures/raggenie | 99dfe5 | Yes | souzatharsis/podcastfy | 804a61 | Yes |
| SpecterOps/cred1py | 432f91 | Yes | StacklokLabs/promptwright | 42f69b | Yes |
| steinathan/reelsmaker | 75369c | Yes | stephengpope/no-code-architects-toolkit | ffc1a8 | Yes |
| StonyBrookNLP/appworld | bc9c47 | Yes | Storia-AI/sage | f47fa4 | No |
| superlinear-ai/raglite | b02c5a | Yes | swarmzero/swarmzero | 6fcd7a | Yes |
| tahnok/colmi_r02_client | 84d3a6 | Yes | taketwo/llm-ollama | dd616e | Yes |
| taobojlen/django-zeal | 232987 | Yes | TencentARC/BrushNet | 101dc3 | No |
| TheAiSingularity/graphrag-local-ollama | bcb98d | Yes | thousandbrainsproject/tbp.monty | a39a26 | Yes |
| THU-MIG/yolov10 | 6fbaf4 | Yes | thu-nics/MoA | da034c | No |
| tjmlabs/AgentRun | 1997dd | Yes | tobiasfshr/map4d | 0b8bcd | Yes |
| Toloka/dbt-af | e7f436 | Yes | TorchJD/torchjd | 1eaafe | Yes |
| tox-dev/tox-uv | d7405a | Yes | TuragaLab/flybody | 2e1088 | Yes |
| turbo-llm/turbo-alignment | 009574 | Yes | TY-Cheng/torchvinecopulib | c3a477 | Yes |
| ucbepic/docetl | 00a761 | Yes | ultrasev/llmproxy | 1a1100 | Yes |
| uname-n/deltabase | 5eafb9 | Yes | unifyai/unify | ea2088 | No |
| Vashkatsi/deply | 6d6875 | Yes | VideoVerses/VideoTuna | ffc6df | Yes |
| vintasoftware/django-ai-assistant | 5b26c7 | Yes | virattt/financial-datasets | 985664 | Yes |
| vllm-project/llm-compressor | 606aab | Yes | volcengine/verl | ed2eaf | Yes |
| volfpeter/fasthx | e850b9 | Yes | volfpeter/htmy | 0322a3 | Yes |
| vysakh0/dravid | 25b03b | Yes | warmshao/FasterLivePortrait | 6aa810 | Yes |
| weareprestatech/hotpdf | 55ab97 | Yes | web-arena-x/visualwebarena | 89f5af | No |
| Weixiang-Sun/Bora | c08bb6 | Yes | whyhow-ai/knowledge-graph-studio | c41043 | Yes |
| whyhow-ai/rule-based-retrieval | 91701f | Yes | whyhow-ai/whyhow | 63a3c6 | Yes |
| WongKinYiu/YOLO | b96c8e | Yes | WU-CVGL/BAD-Gaussians | bdd8b3 | Yes |
| wy-z/container-vm | 07d402 | Yes | xdit-project/xDiT | a7bd51 | Yes |
| xhluca/bm25s | c4fef2 | Yes | yihong0618/klingCreator | e567c6 | Yes |
| yihong1120/Construction-Hazard-Detection | f5e1ca | Yes | yinjunbo/IS-Fusion | 86c882 | No |
| YUCHEN005/GenTranslate | 62e59d | No | YUCHEN005/RobustGER | ad4e37 | No |
| yurujaja/pangaea-bench | e1d12e | Yes | ZenGuard-AI/fast-llm-security-guardrails | 6a867c | Yes |
| zeroasiccorp/logik | ca4bb1 | Yes | zhuqinfeng1999/Samba | 229687 | No |
| zipnn/zipnn | 007319 | Yes | zou-group/textgrad | b2dc68 | Yes |

# I Baseline settings

## I.1 Pipreqs settings

Figure 11 shows the template of a Dockerfile generated using "`requirements_pipreqs.txt`" created by pipreqs.

```dockerfile
FROM python:3.10
WORKDIR /
RUN apt-get update && apt-get install -y curl && \\
    curl -sSL https://install.python-poetry.org | python -
ENV PATH="/root/.local/bin:$PATH"
RUN pip install pytest pytest-xdist && \\
    pip install pipdeptree && \\
    git clone https://github.com/{author_name}/{package_name}.git && \\
    mkdir /repo && \\
    git config --global --add safe.directory /repo && \\
    cp -r {package_name}/. /repo && rm -rf {package_name}/ && \\
    rm -rf {package_name}
COPY requirements_pipreqs.txt /requirements_pipreqs.txt
RUN pip install -r /requirements_pipreqs.txt
RUN cd /repo && pytest --collect-only -q
```

Figure 11: Dockerfile prompt using the "`requirements_pipreqs.txt`" generated by pipreqs.

## I.2 SWE-agent settings

Below is the settings of SWE-agent for building the environment:

```
agent:
  templates:
    system_template: |-
      SETTING: You are an expert skilled in environment configuration,
          operating directly in the command line with a specialized
          interface.
      The specialized interface consists of {{WINDOW}} lines of
          context at a time. In addition to typical bash commands, you
           can also use the following commands to help you with
          environment setup.
      COMMANDS:
      {{command_docs}}
      Please note that THE EDIT COMMAND REQUIRES PROPER INDENTATION.
          Ensure correct indentation when modifying code.
      RESPONSE FORMAT:
      Your shell prompt is formatted as follows:
      (Current task: <task name>) <cwd> $
      You need to format your output using two fields: discussion and
          command.
      Example:
      DISCUSSION
      I'll start by using ls to see the files in the root directory.
      """
      ls /repo
      """
      You should only include a SINGLE command in the command section
          and then wait for a response. Everything in the DISCUSSION
          section will be saved for future reference.
      You can use any other bash commands and the special commands
          listed above, but interactive session commands (e.g., python
          , vim) are not supported.
    instance_template: |-
```

We're currently setting up the environment for the following
    task. Here are the details:
TASK:
{{task_details}}
INSTRUCTIONS:
Now, you'll carry out this task on your own. Your terminal
    session has begun in the repository's root directory. Use
    the provided commands and any bash commands you need. Edit
    and check files as needed.
The goal is to generate a Dockerfile that can successfully build
     and run the tests in the repository using the command "
    pytest --collect-only -q".
NOTE:
1. The repository is cloned into /repo.
2. The Dockerfile should start with the following lines (if the
    base image is python:3.10 and repository is adamobeng/wddbfs
    ):
"""
FROM python:3.10
RUN pip install pytest
RUN git clone https://github.com/adamobeng/wddbfs.git
RUN mkdir /repo
RUN git config --global --add safe.directory /repo
RUN cp -r /wddbfs/. /repo && rm -rf /wddbfs/
RUN rm -rf /wddbfs
"""
Your task includes:
0. **Generate Dockerfile**: Create a file named "Dockerfile" in
    the root path (e.g, the absolute path is /Dockerfile).
1. **Read Directory Structure**: Check the folder structure in
    the root directory.
2. **Check the Configuration Files**: Inspect files like "
    requirements.txt", "setup.py", "setup.cfg", "Pipfile*", etc.
3. **Determine Package Dependencies**: Handle dependencies and
    manage conflicting dependency versions.
4. **Testing**: Ensure "pytest /repo --collect-only -q" runs
    without errors.
5. **Generate Dockerfile**: Write necessary installation or
    setup steps determined from the inspection. If you finish
    run testing successfully, you should modify the Dockerfile
    in /Dockerfile.
IMPORTANT TIPS:
* Check the directory and files carefully.
* Make sure Dockerfile commands are correct.
* Use proper Dockerfile syntax and indentation.
* Test the final Dockerfile by running "pytest /repo --collect-
    only -q".
(Current task: Generate Dockerfile)
(Current directory: {{working_dir}})
bash-$
next_step_template: |-
  {{observation}}
  (Current task: Generate Dockerfile)
  (Current directory: {{working_dir}})
  bash-$
next_step_no_output_template: |-
  Your command ran successfully and did not produce any output.
  (Current task: Generate Dockerfile)
  (Current directory: {{working_dir}})
  bash-$
tools:
  env_variables:
    WINDOW: 100
    OVERLAP: 2
  bundles:

```
      - path: tools/registry
      - path: tools/defaults
      - path: tools/search
      - path: tools/edit_linting
      - path: tools/submit
      - path: tools/env_setup
    parse_function:
      type: thought_action
  history_processors:
    - type: last_n_observations
      n: 5
```

## J    Failure Case

Figure 12 illustrates a "module not found" error when building the repository "`jialuechen/deepfolio`". In this case, the issue arises from the absence of updating the unit tests. The test file "`test_stats.py`" attempts to import modules from "`deepfolio.stats`", but "`stats`" does not exist in "`deepfolio`". Consequently, no matter how the LLM agent operates, it cannot directly run this test. This highlights the importance for developers to continuously update existing unit tests as the code repository evolves.

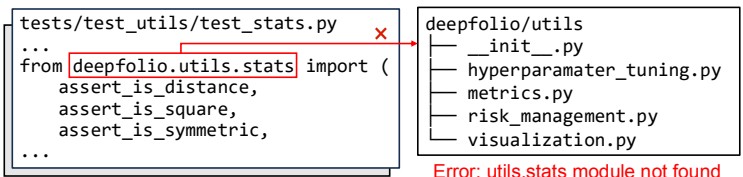

Figure 12: An example of "`module not found`" error due to the absence of updating the unit tests in the repository.