# OpenReview forum: "Repo2Run: Automated Building Executable Environment for Code Repository at Scale"
_NeurIPS.cc/2025/Conference — NeurIPS 2025 spotlight_

### Official Review · Reviewer_q3cZ · 2025-06-23

**Clarity:** 3
**Significance:** 3
**Originality:** 3
**Rating:** 5
**Confidence:** 4

**Summary:**

The paper proposes an agentic solution to automate Dockerfile generation for building executable environment for code repositories at scale. It consists of two phases. In the build phase, the agent interacts with containerized environments to reach a state where all tests are runnable. During this process, the agent can not only execute commands inside the container, but also roll back to a previous container state (checkpointed through *docker commit*), or start over with another base image. In the second record phase, the agent synthesizes a Dockerfile by referencing the command history in the first phase, mapping each command to its corresponding Dockerfile statement step by step. Experiment results show that the proposed Repo2Run method successfully built Dockerfiles of executable environments for 361 out of 420 repositories, significantly outperforming baselines. Further analysis demonstrates the importance of using an external environment for rollback and base image change, and the effectiveness of employing the step-by-step conversion of command history to write the Dockerfile.

**Questions:**

* The two sentences in L265-266 seem duplicated.
* The first sentence in section 6.1 looks abrupt.

**Ethical Concerns:**

["NO or VERY MINOR ethics concerns only"]

**Final Justification:**

I do not have major concerns with the paper and will keep my score.

**Limitations:**

Yes

**Quality:**

4

**Strengths And Weaknesses:**

Strengths:
* Building executable environments has been a key challenge for scalable RLEF (Reinforcement Learning with Execution Feedback) due to the intense human effort required. Repo2Run is a very timely work that addresses the problem by automating it through LLMs. The paper presents very promising results: 86% of the repos can be successfully built and the time cost is comparable to human.
* While the task is relatively new, the experiment setup, including benchmark construction and baseline choices, is sensible.
* The paper is written clearly and easy to follow.

Weaknesses:
* The evaluation only assesses whether all tests are runnable but does not take pass/fail into consideration. The validity of such an approach heavily depends on how the tests are written. For example, if the test files wrap all local imports inside try clauses and handle exceptions as test failures, the agent can then have a shortcut by building an environment only for the test files but not for the source files. To mitigate this, did you consider leveraging human-verified environments (e.g. from SWE-Smith) for evaluation, where the targeted tests are guaranteed to succeed in the groundtruth environment.
* The paper only focuses on Python. While extending the implementation to other programming languages requires a considerable amount of effort and might be out of scope, it is worth having a paragraph discussing the potential challenges for doing so. For example, in the absence of pip, can LLMs figure out the correct package manager based on its own knowledge?

---

> ### Author Rebuttal · Authors · 2025-07-31
>
> We are sincerely grateful for your incredibly supportive and insightful review.
>
> # Response to Weaknesses
>
> ## 1. On the evaluation metric and leveraging human-verified environments.
> This is a very insightful point, and we thank you for raising it. We would like to address the two parts of your suggestion separately.
>
> **On considering test pass/fail rates**
>
> This is an excellent suggestion, and we do consider it deeply during our experimental design. We choose EBSR as our primary metric for several practical reasons:
>
>   (1) For an arbitrary repository, the ground-truth pass rate is unknown. Many have legitimately failing tests, and establishing a reliable baseline for 420 diverse projects would require a massive, subjective manual effort. We lack the personnel to undertake this task. Moreover, it runs counter to our automation goal.
>
>   (2) A failed test can stem from multiple causes, like environment, test bug, source code bug, or wrapping imports in `try...except` clauses, etc. It makes the pass rate a potentially noisy signal for environment quality alone.
>
>   (3) To validate our choice, we conduct a new experiment. We randomly sample 50 successfully built repositories and run the full test suites. This process alone took over 30 hours. The analysis reveals that in these successfully built environments, a majority of tests indeed pass. Specifically, 87.2% of all tests (8,016 out of 9,190) pass, and 36 out of 50 repositories (72%) have a pass rate exceeding 60%. It confirms that EBSR is indeed a proxy for a highly functional and useful environment.
>
> **On leveraging human-verified environments like SWE-Smith**
>
> This is a fantastic idea for future validation. At the time we initiated this work, benchmarks like SWE-Smith were not yet available. Our initial manual experiments on a smaller scale confirm that just getting tests to run is an incredibly time-consuming and difficult task for humans, which motivates our large-scale automated approach. We now see a great opportunity to use curated benchmarks like SWE-Smith as a "gold standard" for evaluating the correctness of our generated environments in future work. Thank you for this suggestion.
>
> ## 2. On discussing generalizability to other languages
>
> This is an excellent point, and we agree that demonstrating generalizability is a critical next step. While we focus on Python due to its prevalence, our framework's design is language-agnostic. It can be adapted to other languages by changing the base image and installation tools, as shown below. We have already implemented a prototype for TypeScript/JavaScript and plan to expand support for more programming languages.
>
> | Language | Docker base image | Installation tool |
> | --- | --- | --- |
> | Python | python:[version] | pip |
> | JavaScript/TypeScript | node:[version] | npm |
> | Java | OpenJDK:[version] | maven |
> | Rust | rust:[version] | cargo |
> | Ruby | ruby:[version] | bundler |
> | R | r-base:[version] | install.packages |
> | Go | golang:[version] | go get |
> | PHP | php:[version] | composer |
>
> We will add a paragraph discussing these potential challenges in the revised version.
>
> # Response to Questions
>
> 1. On the duplicated sentences in L265-266:
> Thank you for your meticulous reading! You are absolutely right; the sentences are redundant. We will correct this in our revised manuscript.
>
> 2. On the abrupt first sentence in Section 6.1:
> Thank you for the helpful suggestion on our writing. We will revise the opening of this section to ensure a smoother transition.
>
> Thank you once again for your positive feedback and the constructive suggestions that have helped us strengthen our paper.

---

> > ### Comment · Reviewer_q3cZ · 2025-08-05
> >
> > Thank you for your response. I do not have further questions and will keep the score.

---

> > > ### Author Response · Authors · 2025-08-06
> > >
> > > Thank you for your support!

---

### Official Review · Reviewer_r9jR · 2025-06-24

**Clarity:** 3
**Significance:** 4
**Originality:** 4
**Rating:** 5
**Confidence:** 5

**Summary:**

The authors propose Repo2Run, an LLM-based agent designed for automatically building and generating Dockerfiles for Python repositories. Their approach entails a dual-environment architecture, with an internal environment corresponding to a Docker container, and an external environment which serves as the bridge between the LLM and internal environment. The external environment is equipped with a "rollback" feature to discard erroneous commands that would otherwise "pollute" the environment. The agent can do many different things like select and change the base image, manage and install dependencies, utilize bash commands, edit files, and execute test suites. Once the agent successfully runs all tests, there is a Dockerfile synthesizer component which converts the command sequence from the building process into a runnable Dockerfile. For evaluation, the authors build a benchmark consisting of 420 Python repositories from GitHub (2024).

**Questions:**

- The underlying LLM for Repo2Run is gpt-4o-2024-05-13. Is the same underlying model used for the "LLM generator" and "SWE-agent" baselines? If it is not, then it seems that much of the difference in performance could be explained due to differences in the base model.
- Due to the rollback mechanism, it seems like the LLM is not given enough information to do self-repair. As a result, do you observe the same erroneous commands being repeatedly generated in subsequent steps, following a rollback?


Suggestions:

- L2: engineerin -> engineering
- The ablation without rollback is not clear. In Table 3, it seems that the only difference between the top 2 rows is the removal of rollback in row #2. In that case, it seems that the performance is actually better without rollback as the numbers in row #2 are higher than row #1. However, I assume the "w/o rollback" includes everything in the full Repo2Run setting including the dual-environment, except for rollback.
- L213-217: Is rollback enabled for the LLM generator? This is a bit confusing.

**Ethical Concerns:**

["NO or VERY MINOR ethics concerns only"]

**Final Justification:**

The authors have addressed the points I raised. I still think this is a good paper that should be accepted. I will keep my score the same.

**Limitations:**

This work only considers Python repositories. There are no empirical results to suggest that this will generalize to other programming languages.

**Quality:**

3

**Strengths And Weaknesses:**

Strengths:
- The authors tackle a very impactful problem — scaling up executable environments for training and evaluating code LLMs is critical for the research community, and the authors have proposed a very novel and clever approach for doing this in an automated manner.
- The findings related to the dual-architecture and rollback mechanism are quite interesting and may be relevant to other tasks for LLM agents.
- The authors have done an excellent job with their analysis. They have extensively covered ablation results, error patterns, performance across different domains, and latency.

Weaknesses:
- The authors have chosen to focus on Python only. It is not clear whether the results will be as high for other languages, as building environments for other languages may be more difficult.

---

> ### Author Rebuttal · Authors · 2025-07-31
>
> We are incredibly grateful for your enthusiastic and insightful review. Thank you for recognizing the impact of our work and the novelty of our approach.
>
> **Response to Questions**
>
> 1. **On the underlying LLM for baselines:**
>
> This is a crucial point for ensuring a fair comparison, and we thank you for raising it. Yes, you are absolutely correct. All LLM-based methods in our paper, including our full Repo2Run agent, the SWE-agent baseline, and the LLM generator baseline, use the exact same underlying model: `gpt-4o-2024-05-13`. We make this deliberate choice to ensure that the significant performance gains demonstrated by Repo2Run stem directly from its superior architectural design and not from an unfair advantage in the base model.
>
> 2. **On the rollback mechanism and self-repair:**
>
> This is a very insightful question that gets to the core of our agent's learning process. Our design of the rollback mechanism is carefully considered. While the mechanism reverts the environment state to a clean checkpoint, **it does not erase the agent's memory**. The full context of the failed attempt, including the command that leads to the error and the resulting error message, is explicitly preserved and fed back into the agent's history.
>
> This ensures the LLM is fully aware of what is wrong, prompting it to learn from the failure and attempt a different, often more effective, strategy (e.g., switching from `pip` to `apt-get`, or changing a dependency version). We do not observe the agent repeatedly making the same mistakes in our experiments, which validates this strategic choice.
>
> **Response to Suggestions**
>
> We sincerely thank you for your meticulous reading and for providing such helpful suggestions.
>
> 1. On the "engineerin" typo:
>
> Thank you for catching this! We will correct it in the revised manuscript.
>
> 2. On the clarity of the ablation study:
>
> Your understanding is correct: the "w/o rollback" ablation removes only the rollback feature while keeping the rest of the dual-environment architecture. We will revise the caption and the corresponding text for Table 3 to make the structure and results of our ablation studies clear and unambiguous.
>
> 3. On rollback for the "LLM generator":
>
> Thank you for seeking this clarification. It is not enabled for any of the baselines, including the "LLM generator". We will make this distinction more explicit in the paper.
>
> **Response to Limitations (Extending to other languages)**
>
> This is an excellent point, and we agree that demonstrating generalizability is a critical next step. While we focus on Python due to its prevalence, our framework's design is language-agnostic. It can be adapted to other languages by changing the Docker base image and installation tools, as shown below. The core structure does not need to change. We have already implemented a prototype for TypeScript/JavaScript and plan to expand support for more programming languages.
> | Language | Docker base image | Installation tool |
> | --- | --- | --- |
> | Python | python:[version] | pip |
> | JavaScript/TypeScript | node:[version] | npm |
> | Java | OpenJDK:[version] | maven |
> | Rust | rust:[version] | cargo |
> | Ruby | ruby:[version] | bundler |
> | R | r-base:[version] | install.packages |
> | Go | golang:[version] | go get |
> | PHP | php:[version] | composer |
>
> We will also add a paragraph discussing this challenge in the revised version.
>
> Thank you once again for your strong support and for your detailed feedback, which has been invaluable to us.

---

### Official Review · Reviewer_S6iE · 2025-06-25

**Clarity:** 3
**Significance:** 3
**Originality:** 3
**Rating:** 4
**Confidence:** 3

**Summary:**

This paper addresses the challenging and practical problem of automating environment configuration for code repositories. It proposes **Repo2Run**, an LLM-driven agent that uses a dual-environment design and rollback mechanism to generate Dockerfiles and build reproducible environments at scale. The authors evaluate their approach on a benchmark of Python repositories, demonstrating strong performance when compared to baselines like SWE‑agent and pipreqs, and report high success in both environment builds (86%) and Dockerfile generation (100%).

**Questions:**

Please respond to the above weaknesses.

**Ethical Concerns:**

["NO or VERY MINOR ethics concerns only"]

**Final Justification:**

I want to keep my original score.

**Limitations:**

Yes

**Quality:**

3

**Strengths And Weaknesses:**

## Strengths

- **Clear motivation & practical significance**: Automating environment setup is a longstanding pain point in software development; framing it as an LLM-based agent problem is timely and relevant to the ML+systems community.
- **Innovative agent architecture**: The dual-environment sandbox plus rollback strategy is a clever solution to mutable build environments and dependency conflicts.
- **Rigorous empirical evaluation**: Experiments span real-world Python repos, include strong baselines, and feature ablation studies—this lends credible support to the core claims.

## Weaknesses

1. **Limited LLM comparisons**: Experiments only use GPT‑4o. It remains unclear whether the method generalizes to other LLMs (e.g., Gemini-2.5, Claude-3.7). Additionally, the LLM‑generator baseline could benefit from swapping in different reasoning models.
2. **Lack of cost/resource analysis**: Although build times (~29 min) are reported, there is no discussion of financial cost.
3. **Reliance on unit tests**: The system depends heavily on existing unit tests for feedback. Many real-world repos lack tests, limiting applicability. No fallback mechanisms are explored.

---

> ### Author Rebuttal · Authors · 2025-07-31
>
> We sincerely thank you for your positive feedback on our work's motivation, innovative architecture, and rigorous evaluation. We appreciate the opportunity to address your constructive questions.
>
> **Regarding Weakness #1: Limited LLM comparisons**
>
> This is an excellent point. To investigate our framework's generalizability, we conduct a new experiment with `Claude 3.7 Sonnet`. Due to time and computational constraints, we randomly sample 50 repositories from our benchmark and replace the base LLM for all LLM-driven methods (Repo2Run, SWE-agent, and LLM generator) with Claude 3.7 Sonnet. The results for both Environment Building Success Rate (EBSR) and Dockerfile Generation Success Rate (DGSR) are presented below:
>
> | Base LLM | Repo2Run | SWE-agent | LLM generator |
> | --- | --- | --- | --- |
> | **EBSR** (GPT-4o) | 80% (40) | 8% (4) | 18% (9) |
> | **EBSR** (Claude 3.7 Sonnet) | 84% (42) | 32% (16) | 32% (16) |
> | **DGSR** (GPT-4o)| 100% (50) | 22.0% (11) | 46% (28) |
> | **DGSR** (Claude 3.7 Sonnet) | 100% (50) | 38% (19) | 64% (32) |
>
> Our analysis of these results is as follows:
>
> As expected, the stronger reasoning capabilities of Claude 3.7 Sonnet lead to a performance increase across all methods. Repo2Run's EBSR improved from 80% to 84%, while SWE-agent and LLM generator see more substantial gains, rising from 8% to 32% and 18% to 32%, respectively.
>
> However, a great performance gap between Repo2Run and the baselines remains. The primary reason for this is architectural. The baselines lack a mechanism to guarantee that the generated Dockerfile is runnable. This is illustrated by the Dockerfile Generation Success Rate (DGSR). While Claude 3.7 Sonnet's improved understanding boosts the DGSR for the baselines, many of their generated Dockerfiles still fail to build. In contrast, Repo2Run's `build-then-record` design ensures a 100% DGSR by definition, as it only synthesizes a Dockerfile from a proven, successful command history.
>
> This confirms that while a more powerful LLM benefits all approaches, the fundamental architectural innovations of Repo2Run are the key drivers of its superior reliability and success.
>
> **Regarding Weakness #2: Lack of cost/resource analysis**
>
> Thank you for this practical suggestion. We follow the precedent set by prior work, like SWE-agent [1], and set a maximum budget of 4 USD per repository. It is widely accepted by the research community. Across all 420 repositories in our benchmark (including both successful and failed attempts), the average cost per repository is approximately 3.56 USD. We believe this is a reasonable and acceptable cost for fully automating such a complex engineering task, especially when compared to the cost of manual human effort. We will add this analysis to the revised version.
>
> **Regarding Weakness #3: Reliance on unit tests**
>
> This is an insightful observation. We choose unit tests as the primary feedback signal for two main reasons:
>
> 1. Unit tests provide the most direct and reliable signal for verifying whether an environment is correctly configured to run the project's code as the original developers intended.
>
> 2. A key application for our work is to enable large-scale training of reinforcement learning models for software engineering (e.g., SWE-RL [2] and SWE-Fixer [3]). In this domain, unit test outcomes are always considered the standard and most trusted form of feedback signal.
>
> While our current focus is on repositories with existing tests, we agree that handling projects without them is also an important future direction. For such cases, one could envision a fallback mechanism where the agent uses static analysis checks as a proxy for success. For example, the agent's goal could be to create an environment where a linter (like `pylint`) runs without import errors, or where a simple script can successfully import the main application modules. Exploring these alternative feedback signals is an avenue for future work, but it is beyond the scope of our paper.
>
> We hope these clarifications, supported by new data and experiments, have fully addressed your concerns. Thank you again for your time and effort.
>
> [1] Yang J, Jimenez C E, Wettig A, et al. SWE-agent: agent-computer interfaces enable automated software engineering[C]//Proceedings of the 38th International Conference on Neural Information Processing Systems. 2024: 50528-50652.
>
> [2] Wei Y, Duchenne O, Copet J, et al. Swe-rl: Advancing llm reasoning via reinforcement learning on open software evolution[J]. arXiv preprint arXiv:2502.18449, 2025.
>
> [3] Xie C, Li B, Gao C, et al. Swe-fixer: Training open-source llms for effective and efficient github issue resolution[J]. arXiv preprint arXiv:2501.05040, 2025.

---

### Official Review · Reviewer_eMpx · 2025-07-01

**Clarity:** 3
**Significance:** 3
**Originality:** 3
**Rating:** 4
**Confidence:** 4

**Summary:**

This paper introduces Repo2Run, an LLM-based agent designed to automate the construction of executable test environments for code repositories and generate runnable Dockerfiles. The approach adopts a dual-environment architecture (internal and external environments) and a Dockerfile synthesizer, and leverages operations such as rollback mechanisms, dependency installation, test running, and code editing. The authors claim that Repo2Run achieves an 86.0% success rate and generates 100% buildable Dockerfiles through evaluation on 420 Python repositories, significantly outperforming existing baselines such as SWE-agent and pipreqs.

**Questions:**

Can the authors provide more detailed arguments for the innovation of the “rollback mechanism” and “dual environment architecture” compared to Docker’s standard practices and the general agent-host control model? Renaming existing technologies does not constitute a true original contribution.

The EBSR metric ignores the actual success or failure of test cases, which makes the definition of “successful build” too loose. Why not use a stricter metric, such as requiring at least a certain percentage of test cases to pass? Among the 361 “successful” build cases, how many cases actually have the majority of tests failing?

Can the authors elaborate on the details of the prompt engineering done when adapting SWE-agent to new tasks? Is it possible that the poor performance of the baseline is caused by poor prompts?

What is ECSR in Tables 1 and 3? Is it EBSR?

**Ethical Concerns:**

["NO or VERY MINOR ethics concerns only"]

**Final Justification:**

The author has not yet addressed all my concerns, so I tend to maintain my original score.

Update:

Although I have raised my score, I wish to articulate my remaining concerns. In my view, the EBSR is not a sufficiently robust metric on its own. A success criterion that only considers whether tests can be executed—regardless of whether they pass or fail—is too permissive and can exaggerate the practical capabilities of the system.

The authors argued that “test pass rates can be a "noisy" signal for environment quality, as failures might stem from bugs in the source code or the tests themselves, not just from build errors”. However, I believe this issue can be mitigated by using human-verified benchmarks where the tests are known to pass in a correctly configured environment.

Nevertheless, I still see some value in this article, so after some hesitation, I chose to slightly lean towards weak accept this article.

**Limitations:**

No discussion of the generalizability of this method.

**Quality:**

2

**Strengths And Weaknesses:**

The paper defines the EBSR as "only caring whether the tests can be executed, regardless of whether they pass or fail". This is an extremely loose success criterion. An "executable environment" that fails all tests due to configuration errors (such as missing environment variables, dependency version conflicts, test data files not found, etc.) cannot be considered a successful build in any practical application scenario. This definition seriously exaggerates the true capabilities of Repo2Run.

The core contribution of this research is over-packaged. The main components of Repo2Run, such as LLM-based agent, Docker container, rollback mechanism, and Dockerfile generator, are not new technical concepts. It combines the application of existing technologies.

The paper admits that its experiments "have not repeated experiments many times to get error bars" because they were time-consuming and expensive, which makes it impossible to judge the stability and contingency of the experimental results, weakening the scientific rigor of the conclusions.

The paper claims that " for complex issues, Repo2Run shows greater advantages over manual building", but does not provide any quantitative data to support this argument.

The experiments rely entirely on the closed-source model gpt-4o-2024-05-13 and are only tested on Python repositories, limiting the generalizability of their conclusions.

---

> ### Author Rebuttal · Authors · 2025-07-31
>
> We appreciate your detailed review and value the opportunity to clarify our design choices and contributions.
>
> # Response to Questions:
>
> **Regarding Question #1: Innovation of "rollback" and "dual environment"**
>
> Our work does not simply rename existing technologies. Our contribution is a novel, specialized architecture for the complex, stateful problem of automated executable environment building, a task for which general-purpose agent models are ill-equipped.
> Our framework is specifically designed for this task, and its core 'build-then-record' paradigm sets it apart from general agent-host models.
>
> Our *dual-environment architecture* is fundamentally different from a general agent-host model. It provides essential meta-level control, most notably through the `base_image_change` action, which allows the agent to switch the entire underlying OS and language version (e.g., from Python 3.10 to 3.6). An agent operating purely inside a container cannot perform such a critical recovery action.
>
> The *rollback mechanism* is more than just `docker commit`; it is a strategic element of the agent's reasoning loop. The agent receives the full history of the failed attempt, allowing it to learn from the error and attempt a more effective sequence on a safe state, rather than getting stuck trying to patch the "polluted" environment.
>
> **Regarding Question #2: EBSR metric**
>
> We chose the EBSR metric after careful consideration of the problem's practical realities.
>
> 1. For any arbitrary "in-the-wild" repository, it is impossible to know the ground-truth test pass rate beforehand. Many repositories have legitimately failing tests as part of their development cycle. Manually establishing a baseline for all 420 repositories would be an immense and subjective undertaking, contrary to our goal of full automation.
>
> 2. A failed test may have three possible root causes: (1) an incorrectly configured environment, (2) a bug in the test code itself, or (3) a bug in the source code that the test is correctly identifying. Simply counting failed tests does not distinguish between these cases and is therefore a noisy signal for environment quality.
>
> 3. Given these factors, we chose EBSR as a practical metric, as it directly measures the agent's ability to create an environment where the developer can begin debugging—the primary goal.
>
> 4. Due to time constraints, we ran the full test suites for 50 successfully built repositories, which took over 30 hours to execute the tests and check the pass number manually. The analysis shows that in 87.2% (8016/9190) of these tests, the majority of tests passed, and 36 repositories (72%) passed over 60% tests. It confirms that EBSR is a strong proxy for a useful environment.
>
> **Regarding Question #3: Prompt engineering for SWE-agent**
>
> We ensure a fair comparison by following the standard practices for SWE-agent:
>
> 1. We use the official SWE-agent framework, which has an established technical precedent for interacting with Docker environments, and its widely adopted prompt template. We only replace the task description from "code fixing" to "executable environment building", providing a clear workflow and examples. We use the `gpt-4o-2024-05-13` model and keep all other settings at default. The exact prompt is available in **Appendix H.2** for full transparency.
>
> 2. We also ensure that the prompts for Repo2Run and SWE-agent are structurally consistent. The performance difference does not stem from "poor prompts" but from fundamental architectural limitations of the baseline. SWE-agent lacks the crucial two-phase build process and specialized actions (like `base image change`) that Repo2Run provides, making it incapable of solving many of the complex setup challenges in our benchmark.
>
> **Regarding Question #4: Typo**
>
> You are correct. This was a typo and should have been EBSR. We apologize for the error and will correct it in the revised manuscript. Thank you for your careful reading.
>
> # Response to Weaknesses:
>
> **On the contribution being "over-packaged"**
>
> We respectfully disagree with this characterization. Our innovation is a complete, specialized framework for a notoriously difficult problem. Simply combining existing technologies would not solve it.
>
> Our framework is purpose-built, with components like the rollback mechanism and Dockerfile synthesizer designed from the ground up for this task. The dual-environment architecture provides essential capabilities, like `base image change`, that are absent in general-purpose agents.
>
> We are encouraged that this view is shared by other reviewers, who described our work as "novel and clever" (R-r9jR) and featuring an "innovative agent architecture" (R-S6iE).
>
> **On the lack of error bars/stability**
>
> To quantitatively address this, we conduct a new experiment on stability. We randomly sample 50 repositories (40 built successfully, 80% EBSR) and rerun the build process two additional times. Repo2Run achieves the same success rate in these reruns. This confirms that once a valid environment path is found, our method is reliable and can deterministically reproduce the result.
>
> **On the lack of quantitative data for advantages over manual building**
>
> Our manual study is detailed in **Appendix E.1**. In that study, human experts successfully configured 26 of 40 repositories (65%), whereas Repo2Run succeeded on 32 of 40 (80%). Notably, Repo2Run solves all cases that the human experts could, plus six more that they could not. The raw data is also available via the anonymous link in our paper.
>
> **On generalizability (LLM and Language)**
>
> 1. **LLM generalizability**: To investigate our framework's generalizability, we conduct a new experiment with Claude 3.7 Sonnet. Due to time and computational constraints, we randomly sample 50 repositories from our benchmark and replace the base LLM for all methods (Repo2Run, SWE-agent, and LLM generator) with `Claude 3.7 Sonnet`. The results for both Environment Building Success Rate (EBSR) and Dockerfile Generation Success Rate (DGSR) are presented below:
>
> | Base LLM | Repo2Run | SWE-agent | LLM generator |
> | --- | --- | --- | --- |
> | **EBSR** (GPT-4o) | 80% (40) | 8% (4) | 18% (9) |
> | **EBSR** (Claude 3.7 Sonnet) | 84% (42) | 32% (16) | 32% (16) |
> | **DGSR** (GPT-4o)| 100% (50) | 22% (11) | 46% (28) |
> | **DGSR** (Claude 3.7 Sonnet) | 100% (50) | 38% (19) | 64% (32) |
>
> Our analysis of these results is as follows:
> As expected, the stronger reasoning capabilities of Claude 3.7 Sonnet lead to a performance increase across all methods. Repo2Run's EBSR improved from 80% to 84%, while SWE-agent and LLM generator see more substantial gains, rising from 8% to 32% and 18% to 32%, respectively.
>
> However, a great performance gap between Repo2Run and the baselines remains. The primary reason for this is architectural. The baselines lack a mechanism to guarantee that the generated Dockerfile is runnable. This is illustrated by the Dockerfile Generation Success Rate (DGSR). While Claude 3.7 Sonnet's improved understanding boosts the DGSR for the baselines, many of their generated Dockerfiles still fail to build. In contrast, Repo2Run's `build-then-record` design ensures a 100% DGSR by definition, as it only synthesizes a Dockerfile from a proven, successful command history.
>
> This confirms that while a more powerful LLM benefits all approaches, the fundamental architectural innovations of Repo2Run are the key drivers of its superior reliability and success.
>
> 2. **Language generalizability**: This is an excellent point, and we agree that demonstrating generalizability is a critical next step. While we focus on Python due to its prevalence, our framework's design is language-agnostic. It can be adapted to other languages by changing the Docker base image and installation tools, as shown below. The core structure does not need to change. We have already implemented a prototype for TypeScript/JavaScript and plan to expand support for more programming languages.
> | Language | Docker base image | Installation tool |
> | --- | --- | --- |
> | Python | python:[version] | pip |
> | JavaScript/TypeScript | node:[version] | npm |
> | Java | OpenJDK:[version] | maven |
> | Rust | rust:[version] | cargo |
> | Ruby | ruby:[version] | bundler |
> | R | r-base:[version] | install.packages |
> | Go | golang:[version] | go get |
> | PHP | php:[version] | composer |
>
> We will also add a paragraph discussing this challenge in the revised version.
>
> We hope these clarifications, supported by new data and experiments, have fully addressed your concerns. Thank you again for your time and effort.

---

> > ### Author Response · Authors · 2025-08-06
> >
> > Dear Reviewer,
> >
> > I hope this message finds you well.
> >
> > As the discussion period is nearing its end, I wanted to follow up on our submission. We have submitted a detailed rebuttal where we believe we have addressed all the questions and weaknesses you raised.
> >
> > We wanted to ensure we have addressed all your concerns satisfactorily. If there are any additional points or feedback you’d like us to consider, please let us know. Your insights are invaluable to us, and we’re eager to address any remaining issues to improve our work.
> >
> > Thank you for your time and effort in reviewing our paper.

---

> ### Comment · Reviewer_eMpx · 2025-08-06
>
> Thank you for your detailed response, but I still have concerns about the statement "EBSR is a strong proxy for a useful environment.", especially since the experiment you used to prove it does not support the claim at all.

---

> > ### Author Response · Authors · 2025-08-06
> >
> > Thank you for your feedback. We have conducted a direct evaluation based on test pass rates.
> >
> > We randomly sample 50 repositories from our benchmark, successfully run their entire test suites, and manually record the pass rates for each method. The table below shows the percentage of repositories for which each method achieved a test pass rate exceeding various thresholds (from >=10% to 100%).
> >
> > | Pass Rate | Repo2Run | SWE-agent | LLM generator | pipreqs |
> > |:----------|:---------|:----------|:--------------|:--------|
> > | >=10%     | **70% (35)** | 26% (13)  | 32% (16)      | 10% (5) |
> > | >=20%     | **68% (34)** | 26% (13)  | 32% (16)      | 10% (5) |
> > | >=30%     | **64% (32)** | 26% (13)  | 32% (16)      | 10% (5) |
> > | >=40%     | **62% (31)** | 26% (13)  | 32% (16)      | 8% (4)  |
> > | >=50%     | **62% (31)** | 24% (12)  | 32% (16)      | 8% (4)  |
> > | >=60%     | **58% (29)** | 20% (10)  | 30% (15)      | 8% (4)  |
> > | >=70%     | **54% (27)** | 18% (9)   | 28% (14)      | 8% (4)  |
> > | >=80%     | **52% (26)** | 18% (9)   | 26% (13)      | 4% (2)  |
> > | >=90%     | **48% (24)** | 18% (9)   | 26% (13)      | 4% (2)  |
> > | 100%      | **34% (17)** | 10% (5)   | 14% (7)       | 4% (2)  |
> >
> > The results are clear: **regardless of the pass rate threshold chosen, Repo2Run consistently and substantially outperforms all baselines.** This experiment demonstrates that even when evaluated by the metric you suggested, our method's superiority remains clear.
> >
> > We conduct this experiment to directly address your concern and show that Repo2Run excels not just in creating executable environments, but in creating high-quality, correct ones.
> >
> > While this new experiment is conducted, we wish to briefly emphasize why we firmly believe that EBSR (Environment Building Success Rate) is an appropriate metric for this task.
> >
> > Our goal is to automate the crucial first step of software usability: creating an executable environment. Test pass rates, while informative, are a "noisy" signal for environment quality because failures can arise from code bugs or test bugs, not just environment building errors. As industry practice (e.g., CI/CD) shows, distinguishing between a build/execution success and a test-pass success is a standard and necessary distinction.
> >
> > We hope this addresses your concerns. We welcome any further questions.

---

> > > ### Author Response · Authors · 2025-08-08
> > >
> > > Dear Reviewer,
> > >
> > > We hope this message finds you well.
> > >
> > > We have provided detailed responses, including results from new experiments using the pass rates as the evaluation metric, to address the concerns you raised.
> > >
> > > As the discussion period is nearing its end, we wanted to ensure our reply has reached you and has sufficiently addressed your questions. Your feedback is very important to us. We would be very grateful if you would consider it in your final assessment and rating of our paper.
> > >
> > > Sincerely,
> > >
> > > The Authors of Submission 5610

---

> > > > ### Comment · Reviewer_eMpx · 2025-08-09
> > > >
> > > > Thanks for your reply. I will update my original score if you can incorporate the changes made during rebuttal into the final version.

---

> > > > > ### Author Response · Authors · 2025-08-09
> > > > >
> > > > > Thank you for your response, and I promise I will incorporate the changes into the final version!

---

### Official Review · Reviewer_EWgw · 2025-07-05

**Clarity:** 3
**Significance:** 3
**Originality:** 3
**Rating:** 4
**Confidence:** 4

**Summary:**

The paper presents a new coding agent framework, Repo2Run, which aims to build executable test environments (e.g., Docker + unit tests) for large-scale code repositories.

Specifically, it involves two phases: the build phase initializes a docker container and runs built-in tests with iterative agent-environment interactions. This phase involves both external and internal environments. The record phase aims to build a dockerfile based on the commands.

**Questions:**

See "weakness" for details.

**Ethical Concerns:**

["NO or VERY MINOR ethics concerns only"]

**Final Justification:**

Thanks for your clarification on the benchmark used and the detailed implementation of dockerfile generation.

Just one clarification, the SUPER dataset has a "dependencies" split (~19.7% of the examples), which requires the agent to install or fix dependencies issues (please see Table 3 in the paper: https://arxiv.org/pdf/2409.07440 for details). I still believe it is directly relevant to your task of "build executable environments".

**Limitations:**

Yes

**Quality:**

2

**Strengths And Weaknesses:**

Strength:
1. Automatic executable environment construction is an important task.
2. The paper presents a pretty decent formulation of the environment construction task, where the goal is to generate a dockerfile for a given repo, and the main evaluation metric is the test success rate.

Weakness:
1. The dockerfile is directly generated by an LLM and would potentially contain malicious code (e.g., commands that take up too much disk space and break the system).  However, when evaluating the dockerfile, it will be directly executed without further examination.
2. The evaluation is mainly on the SWE-Bench dataset with newly introduced metrics. While I agree that both the evaluation setting and the metrics are reasonable, the authors neglect a more relevant benchmark called SUPER: https://arxiv.org/abs/2409.07440. This paper presents the SUPER benchmark that directly evaluates the capability of LLMs in setting up and executing tasks from research repositories, which is more relevant than SWE-Bench.
3. I'm not fully convinced that the authors are comparing to the strongest baselines. For instance, the authors mention OpenHands in the introduction section, but never compare to it, without any further explanations. Since OpenHands is also integrated with a set of execution tools, I'm interested in its performance on environment setup.

---

> ### Author Rebuttal · Authors · 2025-07-31
>
> Thank you for your review. We provide the following clarifications in response to the weaknesses identified.
>
> **Regarding Weakness #1: Dockerfile generation and security**
>
> We would like to begin by correcting a factual misunderstanding. The final Dockerfile is **not** directly generated by the LLM. Our framework's design prevents this:
>
> 1. Our process is two-phased. In the build phase, the LLM's role is to explore and find a valid command sequence within a secure container. In the subsequent record phase, a deterministic **Dockerfile synthesizer** translates this proven success command sequence into the final Dockerfile. By design, commands that take too much disk space and break the system are not included in the synthesizer. The synthesis rule can be found in **Figure 5** and **Appendix B** in our paper. The LLM does not participate in this final synthesis step.
>
> 2. To further validate the quality and safety of our output Dockerfile, we use `hadolint`, a widely-used linter that checks Dockerfiles for syntax errors, security vulnerabilities, and performance issues. As shown in **Table 5** of our paper, Dockerfiles from Repo2Run achieve the lowest warning count, demonstrating they are less prone to common security and performance pitfalls. We are also considering integrating `hadolint` directly into the Repo2Run framework as a quality gate in future work.
>
> **Regarding Weakness #2: The evaluation benchmark and SUPER**
>
> We would like to clarify another factual point: **our evaluation is not conducted on the SWE-Bench dataset.**
>
> 1. We construct our benchmark of 420 repositories to ensure scale and relevance. We are encouraged that our methodology was recognized by other reviewers as "rigorous" (Reviewer-S6iE) and "sensible" (Reviewer-q3cZ).
>
> 2. Furthermore, the SUPER benchmark addresses a fundamentally different task. SUPER's goal is to **reproduce tasks from research papers with curated instructions**, whereas our goal is to **build executable environments for any arbitrary repository**. Our task is designed to solve a more foundational challenge in software engineering and AI for code, and thus involves a more general and complex problem space.
>
> **Regarding Weakness #3: Comparison with OpenHands**
>
> We choose SWE-agent as our primary agent-based baseline for several well-considered reasons.
>
> 1. We choose SWE-agent because it has an established technical precedent for interacting with Docker environments. Through its use in subsequent high-profile work like SWE-smith [1], a clear path for using SWE-agent in Docker-based build and test loops has been demonstrated. This makes it the most direct and scientifically sound baseline to compare against.
>
> 2. In contrast, to the best of our knowledge, OpenHands does not have a readily available, established version that is adapted for this specific `docker build` interaction model. Our contribution is the Repo2Run framework itself, not the separate and significant research effort of adapting a generalist agent to this new domain.
>
> 3. Therefore, comparing against SWE-agent allows for a fair evaluation of our specialized framework against the most relevant existing approach. Forcing a comparison with OpenHands would have required us to first design and build an adaptation for it, which is outside the scope of a baseline comparison.
>
> We hope these clarifications, supported by new data and experiments, have fully addressed your concerns. Thank you again for your time and effort.
>
> [1] Yang J, Leret K, Jimenez C E, et al. Swe-smith: Scaling data for software engineering agents[J]. arXiv preprint arXiv:2504.21798, 2025.

---

> > ### Author Response · Authors · 2025-08-06
> >
> > Dear Reviewer,
> >
> > I hope this message finds you well.
> >
> > As the discussion period is nearing its end, I wanted to follow up on our submission. We have posted a detailed rebuttal and were hoping to ensure we have addressed all your concerns satisfactorily.
> >
> > In our response, we've provided specific clarifications regarding what we believe might be a few misunderstandings, particularly concerning our method for Dockerfile generation and the evaluation benchmarks used. We've also addressed all other questions you raised.
> >
> > Your insights are invaluable to us, and we are very eager to know if our rebuttal has resolved these points. If there are any additional questions or feedback, please let us know. We are ready to further discuss and improve our work.
> >
> > Thank you for your time and effort in reviewing our paper.

---

> > > ### Author Response · Authors · 2025-08-08
> > >
> > > Dear Reviewer,
> > >
> > > We hope this message finds you well.
> > >
> > > We have submitted a detailed response where we sought to address all your concerns and clarify any potential misunderstandings about our methodology.
> > >
> > > As the discussion period is nearing its end, we wanted to ensure our reply has reached you and has sufficiently addressed your questions. Your feedback is very important to us. We would be very grateful if you would consider it in your final assessment and rating of our paper.
> > >
> > > Sincerely,
> > >
> > > The Authors of Submission 5610

---

### Note · Authors · 2025-08-13

Dear PC, SAC, AC, and Reviewers,

We sincerely thank all reviewers for their constructive feedback.

We are pleased with the outcome of our rebuttal. Three reviewers (S6iE, r9jR, and q3cZ) reaffirmed their positive ratings, while another (eMpx) agreed to raise their score based on our new experiments. We believe we have thoroughly addressed every point raised. However, as Reviewer EWgw did not engage in the discussion, we were unable to confirm if our clarifications resolved their initial concerns, which we believe stemmed from factual misunderstandings.

Our work, **Repo2Run**, tackles a foundational challenge for scaling code models: the automated building of executable code environments. It is driven by two core technical innovations:

First, our **Dual-Environment Architecture** provides essential meta-level control (like changing the base image), a capability crucial for solving complex setup failures that general-purpose agents cannot.

Second, our **"Build-then-Record"** strategy ensures reliability. It interactively finds a successful command sequence before deterministically creating a Dockerfile. This two-phase process guarantees a 100% runnable output, fundamentally outperforming direct-generation baselines.

With an 86% success rate on a diverse 420-repository benchmark, Repo2Run delivers a practical and scalable solution. We ensure that all clarifications and new results from our discussion will be reflected in the final paper.

Thank you once again for your hard work and for overseeing this process!

Sincerely,

The Authors of Paper 5610

---

### Decision · Program_Chairs · 2025-09-17

**Decision:**

Accept (spotlight)

**Comment:**

- Summary: This paper proposes Repo2Run, an agentic approach to automatically build test environments for code repositories. The approach involves iteratively building the Docker image of a repository until the agent successfully runs all the tests in the repository in an internal environment as a Docker container. The commands during this phase are then synthesized to an executable Dockerfile for reconstructing the executable environment. The paper also introduces a benchmark composed of repositories from Github with unit tests.
- Strengths:
  - The paper explores a significant problem for training code LLMs enabling further research facilitating training with RLEF. The proposed approach is novel.
  - The paper is well written.
  - Experiments are rigorous and designed well.
- Weaknesses:
  - The generalizability of the method beyond Python remains unknown.
  - The reviewers raised good points around leveraging pass/fail metrics for unit tests, leveraging different models with reasoning capabilities, which were addressed during rebuttal.
  - The proposed approach has heavy dependencies on the existence of unit tests in repositories, which might be limiting.
- Recommendation:
  - All of the five reviewers recommended acceptance of the paper, three of whom gave borderline acceptance scores.
  - I believe this paper will be a good contribution to Neurips 2025. The paper tackles an important research problem enabling further research for training code LLMs and will be of interest to the Neurips community. I recommend its acceptance.